# QUANTUM ALGORITHM FOR FINDING THE NEGATIVE CURVATURE DIRECTION

## ABSTRACT

We present an efficient quantum algorithm aiming to find the negative curvature direction for escaping the saddle point, which is a critical subroutine for many second-order non-convex optimization algorithms. We prove that our algorithm could produce the target state corresponding to the negative curvature direction with query complexity $\tilde{\mathcal{O}}(\text{polylog}(d)\epsilon^{-1})$, where $d$ is the dimension of the optimization function. The quantum negative curvature finding algorithm is exponentially faster than any known classical method which takes time at least $\mathcal{O}(d\epsilon^{-1/2})$. Moreover, we propose an efficient algorithm to achieve the classical read-out of the target state. Our classical read-out algorithm runs exponentially faster on the degree of $d$ than existing counterparts.

## 1 INTRODUCTION

Algorithms for finding the minima of functions have attracted significant attention due in part to their prevalent applications in machine learning, deep learning and robust statistics; in particular, those with good complexity guarantees that can converge to the local minima. Numerous algorithms have been proposed in recent years for finding points that satisfying

$$\|\nabla f(\boldsymbol{x})\| \leq \epsilon_g, \text{and } \lambda_{\min}\left(\nabla^2 f(\boldsymbol{x})\right) \geq -\epsilon_H,$$

where $\epsilon_g, \epsilon_H \in (0,1)$. Recent proposals (Nesterov & Polyak, 2006; Conn et al., 2000; Agarwal et al., 2017) based on second-order Newton-type and first-order methodology have been analyzed from such a perspective. However, those methods normally deal with the situations that the iterations may be trapped in the saddle points, since in many cases, such as deep neural networks (Dauphin et al., 2014; Choromanska et al., 2015), existence of many saddle points is the main bottleneck.

In general non-convex optimization, many algorithms have been proposed to escape the saddle points. These algorithms can be divided into the following two categories: the first-order gradient-based algorithms and the second-order Hessian-based algorithms. Generally, second-order algorithms have better iteration complexity than first-order algorithms (cf. Jin et al. (2017)). However, each iteration in the second-order method involves the computation of the **negative curvature direction**, namely, the eigenvector of a Hessian matrix $\boldsymbol{H} = \nabla^2 f(\boldsymbol{x})$ with negative eigenvalue. This computation could take time $\mathcal{O}(d^2)$ when the Hessian matrix is given, or $\mathcal{O}(d/\sqrt{\epsilon})$ when Lanczos method is used with Gradient information to approximate the Hessian-vector product.

Quantum algorithms have shown great potential to become faster alternatives than classical algorithms for many kinds of problems in the field of linear algebra, including principal component analysis (Lloyd et al., 2014), support-vector machine (Rebentrost et al., 2014), singular value decomposition (Rebentrost et al., 2018). These works encourage us to develop an efficient quantum algorithm for finding the negative curvature. To begin with, we formally define the negative curvature finding problem as follows.

**Negative Curvature Finding (NCF) problem:**  *Given a function $f(\boldsymbol{x}) : \mathbb{R}^d \to \mathbb{R}$ which has L-Lipschitz continuous gradient and the corresponding Hessian , we aim to build a quantum algorithm that could efficiently provide the unit vector $\boldsymbol{u}$ with the condition:*

$$\boldsymbol{u}^T \boldsymbol{H} \boldsymbol{u} \leq -\alpha + \epsilon, \tag{1}$$

*where $0 < \alpha < L$ and $0 < \epsilon < \alpha$; or make the non-vector statement that with high probability there is no unit vector $\mathbf{u}$ satisfying the following condition:*

$$\mathbf{u}^T \mathbf{H} \mathbf{u} < -\alpha. \tag{2}$$

## 1.1 RELATED WORK

Optimization methods for non-convex problems can be roughly divided into first-order and second-order methods, depending on the order of the derivative to the objective function they used. Generally, the second-order methods (Carmon et al., 2018; Agarwal et al., 2017) are exploited to find the effective direction to escape the saddle point. Specifically, finding the Negative Curvature is considered as a critical subroutine to analyze the characteristic of the saddle point.

**First-order algorithms**: For the non-convex problem, the first-order method (Gradient-based method) can find the stationary point, which could be a global minima, local minima or saddle point. However, standard analysis by gradient descent cannot distinguish between saddle points and local minima, leaving open the possibility that gradient descent may get stuck at saddle points. Recently Ge et al. (2015); Jin et al. (2017; 2019) showed that by adding noise at each step, gradient descent can escape all saddle points in a polynomial number of iterations. Lee et al. (2016) proved that under similar conditions, gradient descent with random initialization avoids saddle points even without adding noise. However, each step of Gradient-based methods requires $\mathcal{O}(d)$ operations and their iteration complexity is higher than second-order algorithms (Jin et al., 2017).

**Second-order algorithms**: Traditionally, second-order Newton-based methods can converge to local minima, which use the Hessian information to distinguish between first-order and second-order stationary points. There are two kinds of methods that make use of Hessian information. 1) Hessian-based: trust-region (Conn et al., 2000) and cubic regularization (Nesterov & Polyak, 2006) are two methods, in which the sub-problem is to find the decrease direction based on the given Hessian. The calculation of each iteration involves performing Hessian-vector production, which takes time at least $\mathcal{O}(d^2)$. 2) Hessian-free: The Hessian-free methods use Lanczos method to calculate the negative curvature direction and use gradient to approximate the Hessian-vector product (Agarwal et al., 2017; Carmon et al., 2018; Carmon & Duchi, 2016). The Hessian-free method involves $\mathcal{O}(d\epsilon^{-1/2})$ complexity per iteration. The advantage of the second-order algorithm is the superior iteration complexity than the first-order algorithm. However, using Hessian information usually increases computation time per iteration.

**Quantum Algorithm for Linear Algebra**: There are some proposed quantum algorithms for problems in the related linear algebra field. For example, given copies to quantum state $\rho = \sum_{i,j=1}^{d} x_{ij}|i\rangle\langle j|$, where $x_{ij}$ is the $i,j$-th element of $d \times d$ matrix $X$ with eigen-decomposition $X = \sum_{i=1}^{rank(X)} \lambda_i \mathbf{u}_i \mathbf{u}_i$, previous quantum PCA algorithm (Lloyd et al., 2014) could perform the mapping $\sum_j \beta_j |\mathbf{u}_j\rangle \rightarrow \sum_j \beta_j |\mathbf{u}_j\rangle|\tilde{\lambda}_j\rangle$ in time $\mathcal{O}(\text{polylog}(d)\epsilon^{-3})$. However, the quantum PCA model use the density matrix $\rho$ to store the information of matrix $X$, which implicitly assumes the condition $\|X\|_F = 1$ and $\lambda_{\min}(X)$. Another quantum SVD algorithm (Rebentrost et al., 2018) shows an efficient method to estimate the value $\lambda_j/d$ with error $\epsilon$ in time $\mathcal{O}(\epsilon^{-3})$, for $d \times d$ matrix $X$ with eigenvalues $\{\lambda_j\}_{j=1}^d$. However, this model only suits the case when $\lambda_j/d$ is relatively large, and it would take time $\mathcal{O}(d^3\epsilon^{-3})$ to produce $\epsilon$-estimation on eigenvalues. Moreover, both of these works did not study the classical read-out of the output state, which generally takes time at least $\mathcal{O}(d)$ for $d$-dimensional state (Aaronson, 2015), and could offset the claimed quantum speed-up.

## 1.2 OUR CONTRIBUTION

The contribution of this work can be briefly divided into two parts: 1) an efficient quantum algorithm to generate the required quantum state, which corresponds to the negative curvature direction, and 2) an efficient quantum algorithm to obtain the description of the target state $|\mathbf{u}_t\rangle = \sum_{i=1}^r x_i |\mathbf{s}_i\rangle$, where $\{\mathbf{s}_i\}_{i=1}^r$ is an independent vector set selected from columns of Hessian $\mathbf{H}$ with rank $r$.

**Negative Curvature Finding**: We develop an efficient quantum algorithm to produce the target state $|\mathbf{u}_t\rangle$ (for case (1)) or make the non-vector statement (for case (2)). We provide Proposition 1 as the main result of this part, which guarantees the time complexity of our NCF algorithm:

**Proposition 1.** *There exists a quantum algorithm which could solve the Negative Curvature Finding problem in time $\tilde{\mathcal{O}}(\text{polylog}(d)\text{poly}(r)\epsilon^{-1})$, by providing the target state $|u_t\rangle$ (for case (1)), or making the non-vector statement (for case (2)).*

**Classical Read-out**: The classical read-out problem is one bottleneck for many quantum machine learning algorithms whose results are quantum states. Generally, the read-out of a $d$-dimensional quantum state takes time at least $\mathcal{O}(d)$ (Aaronson, 2015), and could offset the claimed quantum speed-up. In order to solve this dilemma, we develop an efficient quantum algorithm for the classical read-out of the target state. We notice that the target state $|u_t\rangle$ can be written as the linear combination form $|u_t\rangle = \sum_{i=1}^r x_i |s_i\rangle$, where $\{s_i\}_{i=1}^r$ is a linearly independent basis sampled from column vectors $\{h_j\}_{j=1}^d$. The algorithm suits the case when the result quantum state lies in the span of several given states, and may give rise to independent interest.

One advantage of generating the form $|u_t\rangle = \sum_{i=1}^r x_i |s_i\rangle$ is that it provides the target eigenvector as the linear-sum of $r$ columns, which guides the curvature direction in general second-order methods. Our state read-out algorithm contains two subroutines named as the Complete Basis Selection and the State Overlap Estimation. The main results about the Complete Basis Selection and the Classical Read-out are briefly summarized as following theorems:

**Theorem 1.** *There exists a quantum algorithm which takes time $\tilde{\mathcal{O}}(\text{poly}(r)\epsilon^{-2}r^c)$ to find an index set $\{g(i)\}_{i=1}^r$, where $r$ is the rank of $H$, $c = 2\log\frac{4r\|H\|_F}{\epsilon}$ and $\{g(i)\}_{i=1}^r$ forms a complete basis $\{|h_{g(i)}\rangle\}_{i=1}^r$ with probability at least 3/4.*

**Theorem 2.** *The classical description of the target state $|u_t\rangle = \sum_{i=1}^r x_i |s_i\rangle$ could be presented in time $\tilde{\mathcal{O}}(\text{polylog}(d)\text{poly}(r)\epsilon^{-5})$ with error bounds in $\frac{\epsilon}{2}$, when the basis set $\{s_j\}_{j=1}^r$ is given.*

The rest of this paper is organized as follows. Some preliminaries about quantum information are introduced in Section 2. In Section 3, we develop an quantum algorithm to solve the NCF problem. In Section 4, we develop an quantum algorithm which aims to read out the target state. We summarize our results and contributions in Section 5.

## 2    PRELIMINARY

In this section we present some preliminary concepts. Some basic quantum knowledge will be introduced in Section 2.1. Some quantum technics will be introduced in Section 2.2.

### 2.1    NOTATIONS AND DEFINITIONS

In this section, we introduce some useful notations and definitions about quantum computing. The dirac notation is a standard notation in quantum mechanics to describe the quantum states. The form $|x\rangle$ is the state which corresponds to the vector $x$, and the form $\langle y|$ is the state which corresponds to the vector $y^T$. The notation $\langle y|x\rangle$ denotes the value $y^T x/(\|y\|\|x\|)$. The notation $|y\rangle\langle x|$ denotes the matrix $yx^T/(\|y\|\|x\|)$. Quantum state is unitary, which means $\||x\rangle\|^2 = \langle x|x\rangle = 1$. Thus for vector $x \in \mathbb{R}^d$, the state $|x\rangle$ is defined as $\sum_{j=1}^d x_j/\|x\||j\rangle$, where $x_j$ is the $j$-th component of vector $x$ and $\{|j\rangle\}_{j=1}^d$ is the state basis which acts like $\{e_j\}_{j=1}^d$ in classical case. One could obtain information from the quantum state by performing measurement. For example, the measurement of $|x\rangle$ on the basis $\{|j\rangle\}_{j=1}^d$ could randomly produce different index $j$ with probability $x_j^2/\|x\|^2$.

We use $[n]$ to denote the set $\{1, 2, \cdots, n\}$. We denote the norm $\|\cdot\|$ as the $\|\cdot\|_2$ norm for vector and the spectral norm for matrix, if there is no more explanation. $\|A\|_F = (\sum_{i=1}^m \sum_{j=1}^n a_{ij}^2)^{1/2}$ is the Frobenius norm of matrix $A \in \mathbb{R}^{m \times n}$. The lowercase form $h_i$ is defined as the $i$-th column vector of matrix $H \in \mathbb{R}^{d \times d}$. $x_i$ is defined as the $i$-th component of vector $x$. The tensor product of two matrix $A \in \mathbb{R}^{m \times n}$ and $B \in \mathbb{R}^{p \times q}$ is defined as $C = A \otimes B$. The tensor product operation could be performed between vectors, since vector is one special kind of matrix. The tensor product could be defined between quantum states $|x_1\rangle$ and $|x_2\rangle$ which is written as $|x_1\rangle|x_2\rangle$. We present definitions of smoothness and $\gamma$-separation here.

**Definition 1.** (smoothness) *A function $f : \mathbb{R}^d \to \mathbb{R}$ is L-smooth if it has L-Lipschitz continuous gradient, that is $\|\nabla f(x) - \nabla f(y)\| \leq L\|x - y\|, \forall x, y \in \mathcal{X}$, where $\mathcal{X}$ is the domain of $f(x)$.*

**Definition 2.** ($\gamma$-separation) *The set $G = \{a_1, a_2, \cdots, a_n\}$ is said to be $\gamma$-separated if $|a_i - a_j| > \gamma, \forall i, j \in [n]$ and $i \neq j$.*

Based on these definitions, we assume that the Hessian $\boldsymbol{H}$ in this article has two properties:

1. Hessian $\boldsymbol{H} \in \mathbb{R}^{d \times d}$ is a $r$-rank matrix;

2. The absolute value of $\boldsymbol{H}$'s non-zero eigenvalue is $\epsilon$-separated.

The first property is directly derived from the assumption of previous classical non-convex optimization method (Carmon et al., 2018), and the low-rank Hessian case has been observed in neural networks (Gur-Ari et al., 2018). The second property is assumed such that we could distinguish different eigenvalues by their absolute value. We further assume that the Hessian matrix $\boldsymbol{H}$ has the eigen-decomposition $\boldsymbol{H} = \sum_{j=1}^{r} \lambda_j \boldsymbol{u}_j \boldsymbol{u}_j^T$ for the convenience of following discussion.

## 2.2 TECHNIQUES

The motivation idea behind our approach is to perform the quantum singular value estimation model and then generate eigen-states by the post-selection on the output state. Here we introduce two techniques including oracle models and critical conclusions in previous work.

**Quantum Oracle Models** (Kerenidis & Prakash, 2016): For the whole paper, we assume the existence of following quantum oracles, and discuss the query complexity of our algorithms to these oracles. Given Hessian $\boldsymbol{H} \in \mathbb{R}^{d \times d}$, we assume that $\boldsymbol{H}$ is stored in a classical data structure such that the following quantum oracles could be implemented:

$$U_H : |i\rangle|0\rangle \rightarrow |i\rangle|\boldsymbol{h}_i\rangle = \frac{1}{\|\boldsymbol{h}_i\|} \sum_{j=1}^{d} h_{ij}|i\rangle|j\rangle, \forall i \in [d], \tag{3}$$

$$V_H : |0\rangle|j\rangle \rightarrow |\tilde{\boldsymbol{h}}\rangle|j\rangle = \frac{1}{\|\boldsymbol{H}\|_F} \sum_{i=1}^{d} \|\boldsymbol{h}_i\||i\rangle|j\rangle, \forall j \in [d], \tag{4}$$

where $\tilde{\boldsymbol{h}}$ stands for the $d$-dimensional vector whose $i$-th component is $\|\boldsymbol{h}_i\|/\|\boldsymbol{H}\|_F$.

The required data structure has a binary tree form. The sign and square value for each entry are stored in different leaves and the value stored in each parent node is the sum of its children's value. A detail description about this data structure can be referred to (Kerenidis & Prakash, 2016). Denote $T_H$ as the time complexity of these oracles.

**Quantum Singular Value Estimation (SVE)**: Given matrix $\boldsymbol{H} \in \mathbb{R}^{d \times d}$ which has the eigenvalue decomposition $\boldsymbol{H} = \sum_{j=1}^{r} \lambda_j \boldsymbol{u}_j \boldsymbol{u}_j$, previous work (Kerenidis & Prakash, 2016) provided a quantum SVE algorithm, which could be used for estimating singular value or generating eigenstate. Here we briefly introduce their conclusion about the time complexity of their algorithm:

**Theorem 3.** *(Kerenidis & Prakash, 2016) Suppose quantum accesses to oracles (3) and (4) exist. There is an algorithm which could perform the mapping $\sum_j \beta_j |\boldsymbol{u}_j\rangle \rightarrow \sum_j \beta_j |\boldsymbol{u}_j\rangle||\hat{\lambda}_j|\rangle$ with time complexity $\mathcal{O}(T_H \text{polylog}(d)\epsilon^{-1})$, where $\hat{\lambda}_j \in [\lambda_j - \epsilon\|\boldsymbol{H}\|_F, \lambda_j + \epsilon\|\boldsymbol{H}\|_F]$ with probability at least $1 - 1/\text{poly}(d)$.*

# 3 QUANTUM NEGATIVE CURVATURE FINDING ALGORITHM

Our main contribution in this section is the quantum Negative Curvature Finding (quantum NCF) algorithm presented in Algorithm 1. The quantum NCF algorithm solves the NCF problem by providing the target state $|\boldsymbol{u}_t\rangle$ (for case (1)) or making the non-vector statement (for case (2)). The target state $|\boldsymbol{u}_t\rangle$ corresponds to the eigenvector $\boldsymbol{u}_t$ which satisfies the condition $\boldsymbol{u}_t^T \boldsymbol{H} \boldsymbol{u}_t \leq -\alpha + \epsilon/2$. Here we present a tighter restrict on the target state $|\boldsymbol{u}_t\rangle$ to keep a $\epsilon/2$ redundancy for the classical read-out of the quantum state. The quantum NCF Algorithm uses the Proper Eigenvalue Labelling (Algorithm 2) and the Target State Generating Algorithm (Algorithm 5 in Appendix) as subroutines proposed in Section 3.2 and Section 3.3, respectively.

---

**Algorithm 1** Quantum Negative Curvature Finding (Quantum NCF) Algorithm

---

**Input:** Quantum access to oracles $U_H$ and $V_H$. The parameter $\epsilon$ and $\alpha$ in the **NCF problem**.
**Output:** The target state $|u_t\rangle$ whose corrsponding classical unit vector $u_t$ satisfies the condition
$\quad u_t^T H u_t \leq -\alpha + \epsilon/2$; or a statement with high probability that there is no such kind of unit
$\quad$ vector $u$ which satisfies the condition $u^T H u \leq -\alpha$.
1: Label the **proper**(less than $-\alpha + \epsilon/2$) eigenvalue of $H$ (Proper Eigenvalue Labelling).
2: **if** the least eigenvalue of $H$ is less than $-\alpha + \epsilon/2$, **then**
3: $\quad$ generate the target state (Target State Generating Algorithm) and output the state;
4: **else**,
5: $\quad$ claim that there is no such kind of unit vector $u$ which satisfies the condition $u^T H u \leq -\alpha$.
6: **end if**

---

### 3.1 CHALLENGES TO DEVELOP QUANTUM NCF ALGORITHM

The core technical component of our quantum algorithm for the NCF problem is the quantum SVE algorithm. However, there are three major challenges that we have to overcome.

Firstly, the positive-negative eigenvalue problem. In the negative curvature finding problem, we are interested in obtaining eigenvectors with negative eigenvalues. Hence, we cannot directly apply the quantum SVE algorithm since it only gives the estimation on $|\lambda_j|$. In order to overcome this critical issue, we develop Algorithm 4 in Appendix to label negative eigenvalues.

Secondly, since the quantum SVE Algorithm presents $\epsilon$-estimation on singular values with time complexity $\mathcal{O}(T_H \|H\|_F \text{polylog}(d)\epsilon^{-1})$(Theorem 3), we need to provide a tight upper bound for the Frobenius norm $\|H\|_F$, which is shown in Lemma 1:

**Lemma 1.** *Suppose $H \in \mathbb{R}^{d \times d}$ is the Hessian matrix derived from the function $f : \mathbb{R}^d \to \mathbb{R}$ which has the L-Lipschitz continuous gradient. Thus the Frobenius norm of $H$ has the upper bound $\|H\|_F \leq \sqrt{r}L$, where $r$ is the rank of $H$.*

Finally, the input-state problem. For the general superposition state $\sum_j \beta_j |u_j\rangle$, the output state of quantum SVE algorithm has the form $\sum_j \beta_j |u_j\rangle||\hat{\lambda}_j|\rangle$. We could generate different pure state $|u_j\rangle$ with probability $|\beta_j|^2$ by the measurement on eigenvalue register. Thus in order to guarantee a small time complexity, we need to prepare a special input state such that the overlap between the input and the target state is relatively large. We briefly summarize our conclusion on the time complexity of Algorithm 1 in Proposition 2.

**Proposition 2.** *Algorithm 1 takes time $\mathcal{O}(T_H \|H\|_F^5 \text{polylog}(d)\epsilon^{-1})$ to solve the negative curvature finding problem by providing the target state $|u_t\rangle$ or making the statement that there is no unit vector satisfies the condition $u^T H u \leq -\alpha$.*

### 3.2 POSITIVE-NEGATIVE EIGENVALUE DISCRIMINATION

In this section, we propose an algorithm aiming to label the target eigenvalue which is less than $-\alpha + \epsilon/2$. This algorithm helps verifying the existence of solution to the NCF problem and generating the target state. Since the eigenvalue information of $H$ is unknown to us, we need to build the Algorithm 2 to label the **proper** eigenvalue, which would benefit the target state generation task in the following section. The **proper** eigenvalue means the eigenvalue is less than $-\alpha + \epsilon/2$. We view this kind of eigenvalue as our target eigenvalue.

The mean idea of Algorithm 2 is to use the input state $\frac{1}{\|H\|_F} \sum_{j=1}^{r} \lambda_j |u_j\rangle|u_j\rangle$ for the quantum SVE model and obtain the state:

$$\frac{1}{\|H\|_F} \sum_{j=1}^{r} \lambda_j |u_j\rangle|u_j\rangle||\tilde{\lambda}_j|\rangle.$$

The measurement on the eigenvalue register would lead this entangled state collapse to different states $|u_j\rangle|u_j\rangle$ for $j \in [r]$. We could obtain the state $|u_j\rangle$ by neglecting the state in any other register. Using state $|u_j\rangle$ to apply the PNED algorithm (Algorithm 4 in appendix) could provide a discrimination on the positive and negative of the corresponding eigenvalue $\lambda_j$. Thus, we could label the **proper** eigenvalue, for the case that the least eigenvalue is less than $-\alpha + \epsilon/2$; or make the

---

**Algorithm 2** Proper Eigenvalue Labelling

---

**Input:** Quantum access to oracles $U_H$ and $V_H$. The parameter $\epsilon$ and $\alpha$ in the NCF problem.
**Output:** A **proper** label to the singularvalue $|\lambda_j|$ such that $\lambda_j \leq -\alpha + \epsilon/2$ with probability $1 - \delta$,
    or a non-vector statement that there is no unit vector $\boldsymbol{u}$ which satisfies $\boldsymbol{u}^T \boldsymbol{H} \boldsymbol{u} < -\alpha$.

1: **for** $k = 1$ to $\frac{4\|\boldsymbol{H}\|_F^2}{\alpha^2}(2\frac{4\|\boldsymbol{H}\|_F^2}{\alpha^2} \log \frac{1}{\delta} + 3)$ **do**
2:     Create the state $\frac{1}{\|\boldsymbol{H}\|_F} \sum_{j=1}^r \lambda_j |\boldsymbol{u}_j\rangle|\boldsymbol{u}_j\rangle$.
3:     Apply the quantum SVE model to obtain the state $\frac{1}{\|\boldsymbol{H}\|_F} \sum_{j=1}^r \lambda_j |\boldsymbol{u}_j\rangle|\boldsymbol{u}_j\rangle||\tilde{\lambda}_j|\rangle$, where
    $|\tilde{\lambda}_j| \in |\lambda_j| \pm \epsilon/4$ with probability $1 - 1/\text{poly}(d)$.
4:     Measure the eigenvalue register and mark the result.
5:     Use the rest state in the first register as the input to apply the PNED algorithm.
6: **end for**
7: Count the result in step 4 and step 5 to obtain the sequence $\{(|\tilde{\lambda}_j|, n_j, m_j)\}_{j=1}^r$. $n_j$ is the number
    of resulting $|\tilde{\lambda}_j|$ in step 4, and $m_j$ is the number of resulting 1 in step 5 for different $|\tilde{\lambda}_j|$.
8: **if** $\frac{m_j}{n_j} < \frac{1}{2}$ for all $j \in [r]$, **then** make the non-vector statement;
9: **else**, choose the largest $|\tilde{\lambda}_j|$ which satisfies the condition $\frac{m_j}{n_j} > \frac{1}{2}$.
10:     **if** $|\tilde{\lambda}_j| < \alpha - \epsilon/4$, **then** make the no-vector statement;
11:     **else**, label eigenvalue $\lambda_j$ as the **proper** eigenvalue.
12:     **end if**
13: **end if**

---

non-vector statement, for the case that all of eigenvalues are greater than $-\alpha$. We present the time complexity of Algorithm 2 in Theorem 4.

**Theorem 4.** *Algorithm 2 could label the **proper** eigenvalue of $\boldsymbol{H}$ with probability $1 - 1/\text{poly}(d)$, or claim with high probability that there is no unit vector $\boldsymbol{u}$ which satisfies $\boldsymbol{u}^T \boldsymbol{H} \boldsymbol{u} < -\alpha$, with time complexity $\mathcal{O}(T_H \|\boldsymbol{H}\|_F^5 \text{polylog}(d)\epsilon^{-1})$.*

### 3.3 TARGET STATE GENERATING

Suppose the result of Algorithm 2 implies the existence of the target eigenvector $\boldsymbol{u}_t$, which satisfies $\boldsymbol{u}_t^T \boldsymbol{H} \boldsymbol{u}_t \leq -\alpha + \epsilon/2$. Then in order to give a solution to the NCF problem, we need to obtain the vector $\boldsymbol{u}_t$. In this section we construct a quantum procedure to generate the state $|\boldsymbol{u}_t\rangle$. The classical read-out of $|\boldsymbol{u}_t\rangle$, which means to estimate vector $\boldsymbol{u}_t$ from quantum state $|\boldsymbol{u}_t\rangle$, will be discussed in the following section. The idea is very similar to Algorithm 2. We still use state $\frac{1}{\|\boldsymbol{H}\|_F} \sum_{j=1}^r \lambda_j |\boldsymbol{u}_j\rangle|\boldsymbol{u}_j\rangle$ as the input of quantum SVE algorithm to obtain state:

$$\frac{1}{\|\boldsymbol{H}\|_F} \sum_{j=1}^r \lambda_j |\boldsymbol{u}_j\rangle|\boldsymbol{u}_j\rangle||\tilde{\lambda}_j|\rangle.$$

Suppose $\lambda_t$ denotes the eigenvalue of $|\boldsymbol{u}_t\rangle$ that $\lambda_t \leq -\alpha + \epsilon/2$. The measurement on the eigenvalue register could generate state $|\boldsymbol{u}_t\rangle|\boldsymbol{u}_t\rangle$ with probability $P_t = \frac{\lambda_t^2}{\|\boldsymbol{H}\|_F^2} \geq \frac{\alpha^2}{4\|\boldsymbol{H}\|_F^2}$. Thus the probability of generating at least one state $|\boldsymbol{u}_t\rangle$ in $N = \left[4\frac{\|\boldsymbol{H}\|_F^2}{\alpha^2} \log \frac{1}{\delta}\right] + 1$ times of measurement is $1 - (1 - P_t)^N$. There is: $1 - (1 - P_t)^N \geq 1 - e^{-NP_t} \geq 1 - e^{-\log(1/\delta)} = 1 - \delta$. So state $|\boldsymbol{u}_t\rangle$ could be generated in $N$ times of measurement with probability at least $1 - \delta$. By considering the time complexity to run the quantum SVE algorithm ($\mathcal{O}(T_H \|\boldsymbol{H}\|_F \text{polylog}(d)\epsilon^{-1})$) and setting the probability error bound $\delta = 1/\text{poly}(d)$, we could derive the time complexity of generating target state in Theorem 5. The detail of Target State Generating algorithm is in Appendix (Algorithm 5).

**Theorem 5.** *Suppose that Hessian $\boldsymbol{H}$ has the eigenvector $\boldsymbol{u}_t$ with eigenvalue less than $-\alpha + \frac{\epsilon}{2}$. Then state $|\boldsymbol{u}_t\rangle$ could be generated in time $\mathcal{O}(T_H \|\boldsymbol{H}\|_F^3 \text{polylog}(d)\epsilon^{-1})$ with probability at least $1 - 1/\text{poly}(d)$.*

## 4 STATE READ-OUT

In this section, we propose an efficient algorithm to read-out the classical vector $\boldsymbol{u}_t$ from the quantum state $|\boldsymbol{u}_t\rangle$. Generally, the classical read-out of a $d$-dimensional quantum state takes at least $\tilde{\mathcal{O}}(d)$ times of measurement to obtain the form $|\boldsymbol{x}\rangle = \sum_{i=1}^{d} x_i \boldsymbol{e}_i$. Thus the classical read-out of the required state could offset the exponential speed-up (Aaronson, 2015) provided in many quantum machine learning algorithms. In order to avoid this problem, we propose a quantum-classical hybrid procedure to rewrite the target state $|\boldsymbol{u}_t\rangle$ as the linear combination of $r$ states, which is selected from column vectors of Hessian $\boldsymbol{H}$. The main result is stated in Proposition 3.

**Proposition 3.** *The classical description of the target state $|\boldsymbol{u}_t\rangle = \sum_{i=1}^{r} x_i |\boldsymbol{h}_{g(i)}\rangle$[1] could be presented in time $\mathcal{O}(T_H \mathrm{polylog}(d)\mathrm{poly}(r)\epsilon^{-5})$ with error bounds in $\epsilon/2$, when the complete basis set $\{\boldsymbol{h}_{g(j)}\}_{j=1}^{r}$ is given.*

Recall that our Hessian matrix $\boldsymbol{H} \in \mathbb{R}^{d \times d}$ has the eigendecomposition $\boldsymbol{H} = \sum_{j=1}^{r} \lambda_j \boldsymbol{u}_j \boldsymbol{u}_j^T$, where $\lambda_j$ is the eigenvalue of $H$ with respect to eigenvector $\boldsymbol{u}_j$. Then, any eigenvector of $\boldsymbol{H}$ that corresponds to a non-zero eigenvalue could be represented as the linear combination of vectors in $\{\boldsymbol{h}_j\}_{j=1}^{d}$ because $\boldsymbol{H}\boldsymbol{u}_j = \lambda_j \boldsymbol{u}_j$ can be written as $\sum_{i=1}^{d} \boldsymbol{h}_i u_j^{(i)} = \lambda_j \boldsymbol{u}_j$. Since $\boldsymbol{H}$ has the rank of $r$, there exists a subset of complete basis $\{\boldsymbol{h}_{g(i)}\}_{i=1}^{r}$ of the column space, which is sampled from the set $\{\boldsymbol{h}_j\}_{j=1}^{d}$. Thus, any eigenvector $\boldsymbol{u}_j$ could also be represented as the linear combination of vectors in $\{\boldsymbol{h}_{g(i)}\}_{i=1}^{r}$.

Back to the state read-out problem, denote $\boldsymbol{s}_i$ as $\boldsymbol{h}_{g(i)}$ for simplicity. Suppose the target state $|\boldsymbol{u}_t\rangle$ that we generated in the previous section can be written as $|\boldsymbol{u}_t\rangle = \sum_{i=1}^{r} x_i |\boldsymbol{s}_i\rangle$, where $\{x_i\}_{i=1}^{r}$ are coordinates of the state $|\boldsymbol{u}_t\rangle$ under the basis $\{|\boldsymbol{s}_i\rangle\}_{i=1}^{r}$. Thus, instead of simply reading out components of vector $\boldsymbol{u}_t$, we could get the classical description of $|\boldsymbol{u}_t\rangle$ by calculating each $x_i$. Note that the complete basis $\{|\boldsymbol{s}_i\rangle\}_{i=1}^{r}$ is not unique and we only need to identify one of them.

We also notice some recent breakthroughs about quantum-inspired algorithms (Tang, 2019; Arrazola et al., 2019), which are based on sampling technics and FKV Algorithm (Frieze et al., 2004). These quantum-inspired algorithms could perform approximate SVD which also outputs eigenvector as the linear-sum on a group of column vectors. However, in order to cover the whole column space, the quantum-inspired algorithm need to sample at least $\mathcal{O}(\frac{r^2}{\epsilon^2})$ number of columns as basis, while our method generate the linear-sum form exactly on $r$ columns.

### 4.1 COMPLETE BASIS SELECTION

In this section, we develop a quantum algorithm to select a subset $S_I = \{g(1), g(2), \cdots, g(r)\}$ from $[d]$, which corresponds to the complete basis $\{|\boldsymbol{h}_{g(i)}\rangle\}_{i=1}^{r}$. The quantum complete basis selection algorithm can be viewed as the quantum version of Gram-Schmidt orthogonalization: firstly we choose $|\boldsymbol{t}_1\rangle = |\boldsymbol{h}_{g(1)}\rangle$ from the set $\{|\boldsymbol{h}_j\rangle\}_{j=1}^{d}$; then given state set $\{|\boldsymbol{t}_m\rangle\}_{m=1}^{l}$, we choose $|\boldsymbol{t}_{l+1}\rangle \propto |\boldsymbol{h}_{g(l+1)}\rangle - \sum_{m=1}^{l} |\boldsymbol{t}_m\rangle\langle\boldsymbol{t}_m|\boldsymbol{h}_{g(l+1)}\rangle$ from the state set $\{|\boldsymbol{h}_j\rangle - \sum_{m=1}^{l} |\boldsymbol{t}_m\rangle\langle\boldsymbol{t}_m|\boldsymbol{h}_j\rangle\}_{j=1}^{d}$. Since the chosen $|\boldsymbol{t}_{l+1}\rangle$ is orthogonal to states $\{|\boldsymbol{t}_m\rangle\}_{m=1}^{l}$ for each iteration $l \in [r-1]$, the state set $\{|\boldsymbol{t}_m\rangle\}_{m=1}^{r}$ forms an orthonormal basis. Note that the state $|\boldsymbol{t}_m\rangle$ is generated along with an index $g(m)$ for $m \in [r]$, so we would obtain a complete basis index set $\{g(m)\}_{m=1}^{r}$ after the implementation of this quantum algorithm. The quantum complete basis selection algorithm is provided in Algorithm 3, in which we denote:

$$|\phi_1^{(l)}\rangle \equiv \frac{1}{\|\boldsymbol{H}\|_F} \sum_{j=1}^{d} \|\boldsymbol{h}_j\||j\rangle\left\{\left[|\boldsymbol{h}_j\rangle - \sum_{m=1}^{l} |\boldsymbol{t}_m\rangle\langle\boldsymbol{t}_m|\boldsymbol{h}_j\rangle\right]|0\rangle - \sum_{m=1}^{l} |\boldsymbol{t}_m\rangle\langle\boldsymbol{t}_m|\boldsymbol{h}_j\rangle|1\rangle\right\}, \quad (5)$$

as the initial state in each iteration. The time complexity of Algorithm 3 is analyzed in Theorem 6. The detail about Algorithm 3 is in Appendix. We also provide Lemma 2 which provides the time complexity of confirming whether a given set $\{\boldsymbol{s}_i\}_{i=1}^{r}$ are linearly independent.

**Theorem 6.** *The Algorithm 3 takes time $\mathcal{O}(T_H \mathrm{poly}(r)\epsilon^{-2}r^{2\log(4r\|\boldsymbol{H}\|_F/\epsilon)})$ to find an index set $\{g(i)\}_{i=1}^{r}$, which forms a complete basis $\{|\boldsymbol{h}_{g(i)}\rangle\}_{i=1}^{r}$ with probability at least 3/4.*

---

[1]$g(i)$ is the index of the $i$-th column vector in the complete basis.

---

**Algorithm 3** Complete Basis Selection

---

**Input:** Quantum access to oracle $U_H$ and $V_H$.
**Output:** The index set of the complete basis: $S_I = \{g(i)\}_{i=1}^r$.
 1: Initialize the index set $S_I = \varnothing$.
 2: **for** $l = 0$ to $r - 1$ **do**
 3:     Create the state $|\phi_1^{(l)}\rangle$ defined in Equation (5).
 4:     Measure the third register of state $|\phi_1^{(l)}\rangle$ multiple times to get the state $|\phi_2^{(l)}\rangle$ which is proportional to $\frac{1}{\|\boldsymbol{H}\|_F} \sum_{j=1}^d |j\rangle \|\boldsymbol{h}_j\| \left[ |\boldsymbol{h}_j\rangle - \sum_{m=1}^l |\boldsymbol{t}_m\rangle \langle \boldsymbol{t}_m | \boldsymbol{h}_j\rangle \right]$.
 5:     Measure the first register and record the result as $g(l + 1)$.
 6:     Denote $|\boldsymbol{t}_{l+1}\rangle$ as the state proportional to $|\boldsymbol{h}_{g(l+1)}\rangle - \sum_{m=1}^l |\boldsymbol{t}_m\rangle \langle \boldsymbol{t}_m | \boldsymbol{h}_{g(l+1)}\rangle$.
 7:     Update the index set $S_I = S_I \cup \{g(l + 1)\}$.
 8: **end for**

---

**Lemma 2.** *It takes $\mathcal{O}(r^3)$ time to check whether the vector set $\{\boldsymbol{s}_i\}_{i=1}^r$ is linearly independent when the classical access to Hessian $\boldsymbol{H}$ is given, where $\boldsymbol{s}_i$ is sampled from column vectors of matrix $\boldsymbol{H}$.*

### 4.2 Coordinates Estimation

Assume the complete basis $\{|\boldsymbol{s}_1\rangle, |\boldsymbol{s}_2\rangle, \cdots, |\boldsymbol{s}_r\rangle\}$ has been selected out in Section 4.1. Thus the read-out problem could be viewed as solving the equation $|\boldsymbol{u}_t\rangle = \sum_{i=1}^r x_i |\boldsymbol{s}_i\rangle$, where $x_i \in \mathbb{R}$ are unknown variables. The coordinate $\{x_i\}_{i=1}^r$ could be obtained by solving the $r$-dimensional linear equation system $\boldsymbol{Cx} = \boldsymbol{b}$, where $b_i = \langle \boldsymbol{u}_t | \boldsymbol{s}_i\rangle$ and $c_{ij} = \langle \boldsymbol{s}_i | \boldsymbol{s}_j\rangle$ for $i, j \in [r]$. The equation $\boldsymbol{Cx} = \boldsymbol{b}$ could be solved classically in at most $\mathcal{O}(r^3)$ time. Note that we can only get the approximation to $c_{ij}$ or $b_i$ instead of the exact value. Theorem 7 verifies the impact of the approximate error to $c_{ij}$ or $b_i$ on the read-out of the target state.

**Theorem 7.** *Suppose $\tilde{c}_{jk}$ is the $\epsilon_1$-approximation to $c_{jk} = \langle \boldsymbol{s}_j | \boldsymbol{s}_k\rangle$ and $\tilde{b}_j$ is the $\epsilon_2$-approximation to $b_j = \langle \boldsymbol{u}_t | \boldsymbol{s}_j\rangle$, $\forall j, k \in [r]$, where $\epsilon_1 = \frac{\epsilon}{6r^2 \|\boldsymbol{C}^{-1}\|^2}$ and $\epsilon_2 = \frac{\epsilon}{6r \|\boldsymbol{C}^{-1}\|}$. Denote vector $\tilde{\boldsymbol{x}} \in \mathbb{R}^r$ as the solution of $\tilde{\boldsymbol{C}} \boldsymbol{x} = \tilde{\boldsymbol{b}}$. Then $\tilde{\boldsymbol{x}}$ could lead an approximate eigenvector $\tilde{\boldsymbol{u}}_t = \sum_{j=1}^r \tilde{x}_j \boldsymbol{s}_j$, such that $\|\tilde{\boldsymbol{u}}_t - \boldsymbol{u}_t\| \le \epsilon/2$.*

We propose several methods in Appendix to estimate overlap $b_i = \langle \boldsymbol{u}_t | \boldsymbol{s}_i\rangle$ and $c_{ij} = \langle \boldsymbol{s}_i | \boldsymbol{s}_j\rangle$, which are based on the Quantum SWAP Test (Buhrman et al., 2001) and the Hadamard Test. Our proposed quantum algorithms could present $\epsilon_1$-estimation to $c_{ij} = \langle \boldsymbol{s}_i | \boldsymbol{s}_j\rangle$ in time $\mathcal{O}(T_H \epsilon_1^{-2})$ and $\epsilon_2$-estimation to $b_i = \langle \boldsymbol{u}_t | \boldsymbol{s}_i\rangle$ in time $\mathcal{O}((T_{Input} + T_H)\epsilon_2^{-4})$, where $T_{Input}$ is the time to generate state $|\boldsymbol{u}_t\rangle$. Since the time complexity to generate the target state is $\mathcal{O}(T_H \|\boldsymbol{H}\|_F^3 \text{polylog}(d)\epsilon^{-1})$ as proposed in Theorem 5, we could derive Proposition 3.

Considering the time complexity $\mathcal{O}(T_H \|\boldsymbol{H}\|_F^5 \text{polylog}(d)\epsilon^{-1})$ to label the proper eigenvalue and the time complexity $\mathcal{O}(T_H \text{poly}(r)\epsilon^{-2} r^{2 \log(4r\|\boldsymbol{H}\|_F/\epsilon)})$ to generate the complete basis set, we could solve the NCF problem in time $\mathcal{O}(T_H \text{polylog}(d)\text{poly}(r)\epsilon^{-2}(\epsilon^{-3} + r^{2 \log(4r\|\boldsymbol{H}\|_F/\epsilon)}))$ by providing the target vector in the form $\boldsymbol{u}_t = \sum_{i=1}^r x_i \boldsymbol{h}_{g(i)}/\|\boldsymbol{h}_{g(i)}\|$ with error bounded by $\epsilon$ or making the non-vector statement.

## 5 Conclusion

We propose an efficient quantum algorithm for the Negative Curvature Finding problem, which is a critical subroutine in many second-order methods for non-convex optimization. The proposed quantum algorithm could produce the target state in time $\mathcal{O}(T_H \epsilon^{-1} \text{poly}(r)\text{polylog}(d))$ with probability $1 - 1/\text{poly}(d)$, which runs exponentially faster than existing classical methods. Moreover, we propose an efficient hybrid quantum-classical algorithm for the efficient classical read-out of the target state with time complexity $\mathcal{O}(T_H \text{poly}(r)\text{polylog}(d)\epsilon^{-2}(\epsilon^{-3} + r^{2 \log(4r\|\boldsymbol{H}\|_F/\epsilon)}))$, which is exponentially faster on the degree of $d$ than existing general quantum state read-out methods.

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

## A  APPENDIX

### A.1  THE PROOF OF THEOREM 2

*Proof.* The time complexity of Algorithm 1 could be directly obtained by the time complexity of Algorithm 2 and Algorithm 5, whose complexity analysis are presented in Theorem 4 and Theorem 5, respectively. □

### A.2  THE PROOF OF LEMMA 1

*Proof.* Assume $\lambda_1 \leq \lambda_2 \leq \cdots \leq \lambda_d$ are eigenvalues of $\boldsymbol{H}$, we have:

$$\min_{\|\boldsymbol{v}\|=1} \boldsymbol{v}^T \boldsymbol{H} \boldsymbol{v} \leq \lambda_j \leq \max_{\|\boldsymbol{v}\|=1} \boldsymbol{v}^T \boldsymbol{H} \boldsymbol{v}, \ \forall j \in [d].$$

By the definition of the Hessian matrix, for unit vector $\boldsymbol{v}$, we have:

$$\boldsymbol{H}\boldsymbol{v} = \nabla^2 f(\boldsymbol{x})\boldsymbol{v} = \lim_{h \to 0} \frac{\nabla f(\boldsymbol{x}+h\boldsymbol{v}) - \nabla f(\boldsymbol{x})}{h}.$$

From above equation, we can obtain:

$$\boldsymbol{v}^T \boldsymbol{H} \boldsymbol{v} \leq \|\boldsymbol{v}\| \cdot \|\boldsymbol{H}\boldsymbol{v}\| \leq \lim_{h \to 0} \frac{\|\nabla f(\boldsymbol{x}+h\boldsymbol{v}) - \nabla f(\boldsymbol{x})\|}{h} \leq \lim_{h \to 0} \frac{L\|h\boldsymbol{v}\|}{h} = L,$$

and $\boldsymbol{v}^T \boldsymbol{H} \boldsymbol{v} \geq -\|\boldsymbol{v}\| \cdot \|\boldsymbol{H}\boldsymbol{v}\| \geq -\lim_{h \to 0} \frac{\|\nabla f(\boldsymbol{x}+h\boldsymbol{v}) - \nabla f(\boldsymbol{x})\|}{h} \geq -\lim_{h \to 0} \frac{L\|h\boldsymbol{v}\|}{h} = -L.$

Thus, the eigenvalue $\lambda_j$ is bounded in $[-L, L]$ for all $j \in [d]$.

We have $\|\boldsymbol{H}\|_F^2 = \sum_i \sum_j h_{ij}^2 = Tr(\boldsymbol{H} \cdot \boldsymbol{H}) = \sum_j \lambda_j(\boldsymbol{H}^2) \leq rL^2$ , so $\|\boldsymbol{H}\|_F \leq \sqrt{r}L$. □

**Lemma 3.  Hoeffding's inequality (Hoeffding, 1994)**

*Suppose $X_1, X_2, \cdots, X_n$ are independent random variables with bounds $X_i \in [a_i, b_i], \forall i \in [n]$. Define $\overline{X} = \frac{1}{n} \sum_{i=1}^n X_i$ , then $\forall \epsilon > 0$, we have:*

$$P(\overline{X} - E[\overline{X}] \geq \epsilon) \leq \exp\left(-\frac{2n^2\epsilon^2}{\sum_{i=1}^n (b_i - a_i)^2}\right), \tag{6}$$

*and*

$$P(\overline{X} - E[\overline{X}] \leq -\epsilon) \leq \exp\left(-\frac{2n^2\epsilon^2}{\sum_{i=1}^n (b_i - a_i)^2}\right). \tag{7}$$

### A.3  THE DETAIL ABOUT ALGORITHM 4

In Algorithm 4, $\boldsymbol{P} \in \mathbb{R}^{d^2 \times d}$ is the matrix whose column vector $\boldsymbol{p}_i = \boldsymbol{e}_i \otimes \frac{\boldsymbol{h}_i}{\|\boldsymbol{h}_i\|}$ for $i \in [d]$ , and $\boldsymbol{Q} \in \mathbb{R}^{d^2 \times d}$ is the matrix whose column vector $\boldsymbol{q}_j = \frac{\tilde{\boldsymbol{h}}}{\|\boldsymbol{H}\|_F} \otimes \boldsymbol{e}_j$ for $j \in [d]$. $\boldsymbol{h}_i$ is the $i$-th column vector of matrix $\boldsymbol{H}$ and $\tilde{\boldsymbol{h}}$ is a $d$-dimensional vector whose $i$-th component is $\|\boldsymbol{h}_i\|$. It can be directly obtained that the matrix $\boldsymbol{P}$ and $\boldsymbol{Q}$ satisfy the decomposition $\boldsymbol{H}/\|\boldsymbol{H}\|_F = \boldsymbol{P}^T \boldsymbol{Q}$ and have property $\boldsymbol{P}^T \boldsymbol{P} = \boldsymbol{Q}^T \boldsymbol{Q} = I$. Mappings $|\boldsymbol{x}\rangle|0\rangle \rightarrow |\boldsymbol{P}\boldsymbol{x}\rangle$ and $|0\rangle|\boldsymbol{x}\rangle \rightarrow |\boldsymbol{Q}\boldsymbol{x}\rangle$ can be performed by the quantum oracle $U_H$ and $V_H$ respectively.

---

**Algorithm 4** Positive-Negative Eigenvalue Discrimination(PNED) Algorithm

---

**Input:** Quantum access to oracles $U_H$ and $V_H$. Eigenstate $|\boldsymbol{u}\rangle$ which corresponds to eigenvalue $\lambda$.
**Output:** A random boolean variable with distribution $P(0) = \frac{1+\lambda/\|\boldsymbol{H}\|_F}{2}$ and $P(1) = \frac{1-\lambda/\|\boldsymbol{H}\|_F}{2}$.
  1: Create state $|\boldsymbol{u}\rangle|0\rangle|0\rangle$.
  2: Apply the Hadmard gate on the third register to obtain the state $\frac{1}{\sqrt{2}}(|\boldsymbol{u}\rangle|0\rangle|0\rangle + |\boldsymbol{u}\rangle|0\rangle|1\rangle)$.
  3: Apply the Controlled-SWAP gate to obtain the state $\frac{1}{\sqrt{2}}(|\boldsymbol{u}\rangle|0\rangle|0\rangle + |0\rangle|\boldsymbol{u}\rangle|1\rangle)$.
  4: Apply gate $U_H \otimes |0\rangle\langle 0| + V_H \otimes |1\rangle\langle 1|$ on the state to obtain $\frac{1}{\sqrt{2}}(|\boldsymbol{P}\boldsymbol{u}\rangle|0\rangle + |\boldsymbol{Q}\boldsymbol{u}\rangle|1\rangle)$.
  5: Apply the Hadmard gate on the third register to obtain the state $\frac{|\boldsymbol{P}\boldsymbol{u}\rangle+|\boldsymbol{Q}\boldsymbol{u}\rangle}{2}|0\rangle + \frac{|\boldsymbol{P}\boldsymbol{u}\rangle-|\boldsymbol{Q}\boldsymbol{u}\rangle}{2}|1\rangle$.
  6: Measure the third register and output the result.

---

**Theorem 8. Positive-negative eigenvalue discrimination.** *For eigenvalue $\lambda$ of $\boldsymbol{H}$ with property $|\lambda| \geq a$, one could run Algorithm 4 for $n = 2[\frac{\|\boldsymbol{H}\|_F^2}{a^2} \log \frac{1}{\delta} - \frac{1}{2}] + 3$ times, to make a statement that $\lambda$ is positive or negative, with probability $1 - \delta$.*

*Proof.* The measurement in step 6 of Algorithm 4 outputs 1 with probability:

$$P(1) = \|\frac{|\boldsymbol{P}\boldsymbol{u}\rangle - |\boldsymbol{Q}\boldsymbol{u}\rangle}{2}\|^2 = \frac{1}{4}((\langle\boldsymbol{P}\boldsymbol{u}| - \langle\boldsymbol{Q}\boldsymbol{u}|)(|\boldsymbol{P}\boldsymbol{u}\rangle - |\boldsymbol{Q}\boldsymbol{u}\rangle)). \tag{8}$$

Note that $\langle\boldsymbol{P}\boldsymbol{u}|\boldsymbol{Q}\boldsymbol{u}\rangle = \boldsymbol{u}^T\boldsymbol{P}^T\boldsymbol{Q}\boldsymbol{u} = \frac{1}{\|\boldsymbol{H}\|_F}\boldsymbol{u}^T\boldsymbol{H}\boldsymbol{u} = \frac{\lambda}{\|\boldsymbol{H}\|_F}$, so $\boldsymbol{P}(1) = \frac{1-\lambda/\|\boldsymbol{H}\|_F}{2}$ . Similarly we have $\boldsymbol{P}(0) = \frac{1+\lambda/\|\boldsymbol{H}\|_F}{2}$.

Suppose that we need $2x + 1$ times of measurement to give an $1 - \delta$ correct statement about whether $\lambda > 0$ or $\lambda < 0$. The problem can be viewed as the biased coin problem. Define random variables $X_i$ such that $\boldsymbol{P}(X_i = 1) = p$ and $\boldsymbol{P}(X_i = 0) = 1 - p$ and $S_n = \sum_{i=1}^n X_i$. Then there has the Hoeffding's inequality $\boldsymbol{P}(S_n/n - p \leq -\epsilon) \leq e^{-2n\epsilon^2}$ and $\boldsymbol{P}(S_n/n - p \geq \epsilon) \leq e^{-2n\epsilon^2}$.

Back to the problem, suppose $\lambda < 0$, by setting $n = 2x + 1$, $n(p - \epsilon) = x$ and $\boldsymbol{P} = \frac{1-\lambda/\|\boldsymbol{H}\|_F}{2}$, we have:

$$P(S_{2x+1} \leq x) \leq \exp(-2(2x+1)[\frac{1 - \lambda/\|\boldsymbol{H}\|_F}{2} - \frac{x}{2x+1}]^2) < e^{-\frac{2x+1}{2}\frac{\lambda^2}{\|\boldsymbol{H}\|_F^2}} \leq e^{-\frac{2x+1}{2}\frac{a^2}{\|\boldsymbol{H}\|_F^2}}.$$

Similarly for $\lambda > 0$, there is $P(S_{2x+1} \geq x) \leq e^{-\frac{2x+1}{2}\frac{a^2}{\|\boldsymbol{H}\|_F^2}}$.

Let $e^{-\frac{2x+1}{2}\frac{a^2}{\|\boldsymbol{H}\|_F^2}} \leq \delta$ , we have $x \geq [\frac{\|\boldsymbol{H}\|_F^2}{a^2} \log \frac{1}{\delta} - \frac{1}{2}] + 1$. $\qquad\square$

### A.4 THE PROOF OF THEOREM 4

*Proof.* The input state $\frac{1}{\|\boldsymbol{H}\|_F} \sum_{j=1}^r \lambda_j|\boldsymbol{u}_j\rangle|\boldsymbol{u}_j\rangle$ could be generated with oracles $U_H$ and $V_H$:

$$|0\rangle|0\rangle \xrightarrow{V_H} \frac{1}{\|\boldsymbol{H}\|_F} \sum_{i=1}^d \|\boldsymbol{h}_i\||i\rangle|0\rangle \xrightarrow{U_H} \frac{1}{\|\boldsymbol{H}\|_F} \sum_{i=1}^d \sum_{j=1}^d h_{ij}|i\rangle|j\rangle. \tag{9}$$

Since $\boldsymbol{H}$ has the eigen-decomposition $\boldsymbol{H} = \sum_{k=1}^r \lambda_k \boldsymbol{u}_k \boldsymbol{u}_k^T$, we could rewrite entry $h_{ij}$ as $h_{ij} = \sum_{k=1}^r \lambda_k u_k^{(i)} u_k^{(j)}$, where $u_k^{(i)}$ is the $i$-th component of vector $\boldsymbol{u}_k$. Thus the state $\frac{1}{\|\boldsymbol{H}\|_F} \sum_{i=1}^d \sum_{j=1}^d h_{ij}|i\rangle|j\rangle$ could be written as:

$$\frac{1}{\|\boldsymbol{H}\|_F} \sum_{i=1}^d \sum_{j=1}^d \sum_{k=1}^r \lambda_k u_k^{(i)} u_k^{(j)}|i\rangle|j\rangle = \frac{1}{\|\boldsymbol{H}\|_F} \sum_{k=1}^r \lambda_k|\boldsymbol{u}_k\rangle|\boldsymbol{u}_k\rangle.$$

Then we apply the quantum SVE model on this state. In order to give $\epsilon/4$-estimation on the singular value, the time complexity to run the quantum SVE algorithm is $\mathcal{O}(T_H\|\boldsymbol{H}\|_F \text{polylog}(d)\epsilon^{-1})$ by Theorem 3.

Suppose there are eigenvalues $\lambda_j$ which are less than $-\alpha + \epsilon/2$. We denote the least one as $\lambda_t$ and label it as the **proper** eigenvalue. By Theorem 8, we need to generate $n_t = 2[\frac{\|\boldsymbol{H}\|_F^2}{\lambda_t^2} \log \frac{1}{\delta} - \frac{1}{2}] + 3$ numbers of state $|\boldsymbol{u}_t\rangle$ in order to guarantee that $\lambda_t < 0$ with probability $1 - \delta$. Note that the probability of generating state $|\boldsymbol{u}_t\rangle$ in each iteration of step 2-4 in Algorithm 2 is $P_t = \frac{\lambda_t^2}{\|\boldsymbol{H}\|_F^2}$. So averagely we need to perform step 2-4 in Algorithm 2 for $n = \frac{\|\boldsymbol{H}\|_F^2}{\lambda_t^2}\{2[\frac{\|\boldsymbol{H}\|_F^2}{\lambda_t^2} \log \frac{1}{\delta} - \frac{1}{2}] + 3\}$ times. The number $n$ can be roughly upper bounded by $\frac{4\|\boldsymbol{H}\|_F^2}{\alpha^2}(2\frac{4\|\boldsymbol{H}\|_F^2}{\alpha^2} \log \frac{1}{\delta} + 3)$, since for negative curvature case $\epsilon < \alpha$, we have $|\lambda_t| = \alpha - \epsilon/2 > \alpha/2$.

By considering the time complexity to run the quantum SVE algorithm $(\mathcal{O}(T_H\|\boldsymbol{H}\|_F \text{polylog}(d)\epsilon^{-1}))$ and setting the probability error bound $\delta = 1/\text{poly}(d)$, we could derive that the time complexity of Algorithm 2 is $\mathcal{O}(T_H\|\boldsymbol{H}\|_F^5 \text{polylog}(d)\epsilon^{-1})$. □

## A.5 TARGET STATE GENERATING ALGORITHM

---

**Algorithm 5** Target State Generating

---

**Input:** Quantum access to oracles $U_H$ and $V_H$. Parameters $\alpha$ and $\epsilon$ in the **NCF problem**.
**Output:** The target state $|\boldsymbol{u}_t\rangle$ with property $\langle \boldsymbol{u}_t|H|\boldsymbol{u}_t\rangle = \lambda_t \leq -\alpha + \epsilon/2$.
 1: **for** $k = 1$ to $\left[4\frac{\|\boldsymbol{H}\|_F^2}{\alpha^2} \log \frac{1}{\delta}\right] + 1$ **do**
 2:     Create the state $\frac{1}{\|\boldsymbol{H}\|_F} \sum_{j=1}^r \lambda_j|\boldsymbol{u}_j\rangle|\boldsymbol{u}_j\rangle$.
 3:     Apply the quantum SVE model to obtain the state $\frac{1}{\|\boldsymbol{H}\|_F} \sum_{j=1}^r \lambda_j|\boldsymbol{u}_j\rangle|\boldsymbol{u}_j\rangle||\tilde{\lambda}_j|\rangle$, where $|\tilde{\lambda}_j| \in |\lambda_j| \pm \epsilon/4$ with probability $1 - 1/\text{poly}(d)$.
 4:     Measure the eigenvalue register and mark the result.
 5:     **if** the eigenvalue measured in in step 4 is labelled to be **proper** in Algorithm 2, **then**
 6:         output the state in the first register as the target state.
 7:     **end if**
 8: **end for**

---

## A.6 THE DETAIL ABOUT ALGORITHM 3

Consider the state:

$$\left[\prod_{j=1}^l |g(j)\rangle\right] \otimes |\phi_1^{(l)}\rangle \equiv \frac{1}{\|\boldsymbol{H}\|_F}\left[\prod_{j=1}^l |g(j)\rangle\right] \sum_{j=1}^d \|\boldsymbol{h}_j\||j\rangle \left\{\left[|\boldsymbol{h}_j\rangle - \sum_{m=1}^l |\boldsymbol{t}_m\rangle\langle \boldsymbol{t}_m|\boldsymbol{h}_j\rangle\right]|0\rangle - \sum_{m=1}^l |\boldsymbol{t}_m\rangle\langle \boldsymbol{t}_m|\boldsymbol{h}_j\rangle|1\rangle\right\},$$
(10)

we note that the state $|\phi_1^{(l)}\rangle$ in Step 3 of Algorithm 3 can be generated by the following procedure:

$$\left[\prod_{j=1}^{l}|g(j)\rangle\right]|0\rangle|0\rangle \xrightarrow{(a)} \frac{1}{\|\boldsymbol{H}\|_F}\left[\prod_{j=1}^{l}|g(j)\rangle\right]\sum_{j=1}^{d}\|\boldsymbol{h}_j\|\,|j\rangle|\boldsymbol{h}_j\rangle \tag{11}$$

$$\xrightarrow{(b)} \frac{1}{\|\boldsymbol{H}\|_F}\left[\prod_{j=1}^{l}|g(j)\rangle\right]\sum_{j=1}^{d}\|\boldsymbol{h}_j\|\,|j\rangle|\boldsymbol{h}_j\rangle\frac{|0\rangle+|1\rangle}{\sqrt{2}} \tag{12}$$

$$\xrightarrow{(c)} \frac{1}{\|\boldsymbol{H}\|_F}\left[\prod_{j=1}^{l}|g(j)\rangle\right]\sum_{j=1}^{d}\|\boldsymbol{h}_j\|\,|j\rangle\left\{\left[|\boldsymbol{h}_j\rangle-2\sum_{m=1}^{l}|\boldsymbol{t}_m\rangle\langle\boldsymbol{t}_m|\boldsymbol{h}_j\rangle\right]\frac{|0\rangle}{\sqrt{2}}+|\boldsymbol{h}_j\rangle\frac{|1\rangle}{\sqrt{2}}\right\} \tag{13}$$

$$\xrightarrow{(d)} \frac{1}{\|\boldsymbol{H}\|_F}\left[\prod_{j=1}^{l}|g(j)\rangle\right]\sum_{j=1}^{d}\|\boldsymbol{h}_j\|\,|j\rangle\left\{\left[|\boldsymbol{h}_j\rangle-\sum_{m=1}^{l}|\boldsymbol{t}_m\rangle\langle\boldsymbol{t}_m|\boldsymbol{h}_j\rangle\right]|0\rangle-\sum_{m=1}^{l}|\boldsymbol{t}_m\rangle\langle\boldsymbol{t}_m|\boldsymbol{h}_j\rangle|1\rangle\right\} \tag{14}$$

$$\xrightarrow{(e)} \frac{1}{\|\boldsymbol{H}\|_F}\sum_{j=1}^{d}\|\boldsymbol{h}_j\|\,|j\rangle\left\{\left[|\boldsymbol{h}_j\rangle-\sum_{m=1}^{l}|\boldsymbol{t}_m\rangle\langle\boldsymbol{t}_m|\boldsymbol{h}_j\rangle\right]|0\rangle-\sum_{m=1}^{l}|\boldsymbol{t}_m\rangle\langle\boldsymbol{t}_m|\boldsymbol{h}_j\rangle|1\rangle\right\}. \tag{15}$$

In step (a), we employ the quantum oracles $U_H V_H$ on ancillas $|0\rangle|0\rangle$. In step (b), an auxiliary register $|0\rangle$ is appended, followed by a Hadamard gate. In step (c), we apply operation $\prod_{m=1}^{l}[R_m\otimes |0\rangle\langle 0|+I\otimes|1\rangle\langle 1|]$. The operation $R_m$ is defined on the register $|g(1)\rangle,\cdots|g(m)\rangle,|\boldsymbol{h}_j\rangle$:

$$R_m = I^{\otimes(m+1)} - 2\left\{\left[\prod_{j=1}^{m}|g(j)\rangle\right]|\boldsymbol{t}_m\rangle\right\}\left\{\left[\prod_{j=1}^{m}\langle g(j)|\right]\langle\boldsymbol{t}_m|\right\}.$$

In step (d), we apply the Hadamard gate on the auxiliary register. In step (e), we trace out registers $\prod_{j=1}^{l}|g(j)\rangle$ to generate state $|\phi_1^{(l)}\rangle$.

The crucial part in Algorithm 3 is to implement the reflection $R_m$. For the $m+1$ case, there is:

$$|\boldsymbol{t}_{m+1}\rangle = \frac{1}{Z_{m+1}}(|\boldsymbol{s}_{m+1}\rangle - \sum_{i=1}^{m}|\boldsymbol{t}_i\rangle\langle\boldsymbol{t}_i|\boldsymbol{s}_{m+1}\rangle), \tag{16}$$

Note that $\{|\boldsymbol{t}_i\rangle\}$ forms the orthogonal basis: $\langle\boldsymbol{t}_i|\boldsymbol{t}_j\rangle = 0, \forall i\neq j$, and $Z_{m+1} = \langle\boldsymbol{t}_{m+1}|\boldsymbol{s}_{m+1}\rangle = \||\boldsymbol{s}_{m+1}\rangle-\sum_{i=1}^{m}|\boldsymbol{t}_i\rangle\langle\boldsymbol{t}_i|\boldsymbol{s}_{m+1}\rangle\| = \sqrt{1-\sum_{i=1}^{m}\langle\boldsymbol{t}_i|\boldsymbol{s}_{m+1}\rangle^2}$.

Denote $|\boldsymbol{t}_i\rangle \equiv \sum_{j=1}^{i}x_{ij}|\boldsymbol{s}_j\rangle$. The restriction that $|\boldsymbol{t}_{m+1}\rangle$ is normalized and orthogonal to states $|\boldsymbol{s}_1\rangle,|\boldsymbol{s}_2\rangle,\cdots|\boldsymbol{s}_m\rangle$ could yield Equation (17):

$$\begin{cases} \sum_{i=1}^{m+1}x_{m+1,i}\langle\boldsymbol{s}_j|\boldsymbol{s}_i\rangle = 0, \ \forall j\in[m], \\ \sum_{j=1}^{m+1}\sum_{i=1}^{m+1}x_{m+1,j}x_{m+1,i}\langle\boldsymbol{s}_j|\boldsymbol{s}_i\rangle = 1. \end{cases} \tag{17}$$

Note that $x_{m+1,m+1} = 1/Z_{m+1}$ by Equation (16). Define the $m$-dimensional vectors $\boldsymbol{x},\boldsymbol{b}$, and the $m\times m$ matrix $\boldsymbol{C}_m$ to be:

$$\boldsymbol{x} = \sum_{i=1}^{m}x_{m+1,i}\boldsymbol{e}_i \ , \boldsymbol{b} = \sum_{j=1}^{m}\langle\boldsymbol{s}_j|\boldsymbol{s}_{m+1}\rangle\boldsymbol{e}_j \ , \boldsymbol{C}_m = \{c_{ij}\equiv\langle\boldsymbol{s}_i|\boldsymbol{s}_j\rangle\}_{i,j}^{m,m},$$

where each value $c_{ij} = \langle\boldsymbol{s}_i|\boldsymbol{s}_j\rangle$ could be estimated by Hadamard Test. Thus, we could derive the following linear equations about $\boldsymbol{x}$ from Equation (17):

$$\begin{cases} \boldsymbol{C}_m\boldsymbol{x} = -\dfrac{1}{Z_{m+1}}\boldsymbol{b}, \\ \boldsymbol{x}^T\boldsymbol{b} = \dfrac{Z_{m+1}^2-1}{Z_{m+1}}. \end{cases} \tag{18}$$

We could obtain the coordinate $\{x_{m+1,i}\}_{i=1}^{m+1}$ by solving Equation (18). There is:

$$
\begin{cases}
x_{m+1,m+1} = \dfrac{1}{Z_{m+1}} = \sqrt{\dfrac{|C_m|}{|C_{m+1}|}}, \\[4mm]
x_{m+1,i} = -\dfrac{1}{Z_{m+1}} \dfrac{|C_m^{(i)}|}{|C_m|}, \quad \forall i \in [m],
\end{cases}
\tag{19}
$$

where matrix $C_m^{(i)}$ denotes the matrix generated from $C_m$ by replacing the $i$-th column with $b$.

Suppose now we have obtained the linear combination form $|t_{m+1}\rangle = \sum_{i=1}^{m+1} x_{m+1,i}|s_i\rangle$. Thus, we could construct a unitary to generate the state $\left[\prod_{j=1}^{m+1}|g(j)\rangle\right]|t_{m+1}\rangle$ from the state $\left[\prod_{j=1}^{m+1}|g(j)\rangle\right]|0\rangle$, by the linear-combination-of-states method (Shao, 2018).

The idea of linear combination of states was introduced in Shao (2018), which focuses on the following problem: given quantum states $|a\rangle$ and $|b\rangle$, to prepare the state $|c\rangle = \frac{1}{Z_c}(x|a\rangle + y|b\rangle)$. The method is based on the fact that $R_{2\theta} = (I - 2|b\rangle\langle b|)(I - 2|a\rangle\langle a|)$ can be viewed as the clockwise rotation in the plane spanned by $|a\rangle$ and $|b\rangle$ with angle $2\theta$, where $\theta = \arccos\langle a|b\rangle$ is the angle between $|a\rangle$ and $|b\rangle$. Thus any clockwise rotation in space $SPAN\{|a\rangle, |b\rangle\}$ with angle $\phi$ could be written as $R_\phi = R_{2\theta}^{\phi/2\theta}$. For the case $|c\rangle = \frac{1}{Z_c}(x|a\rangle + y|b\rangle)$, there is $|c\rangle = R_\phi|a\rangle$, where $\phi = \arccos\frac{x+y\langle a|b\rangle}{Z_c} = \arccos\frac{x+y\langle a|b\rangle}{\sqrt{x^2+y^2+2xy\langle a|b\rangle}}$. The linear sum of 2 states could be generalized to $n$ case:

**Theorem 9.** (Shao, 2018) *Assume state $|\phi_i\rangle$ could be prepared by given unitary operation in time $T_{in}$, for $i \in [n]$. Then there is a unitary which could prepare the state $|\phi\rangle = \sum_{i=1}^{n} \alpha_i|\phi_i\rangle$ in time $\mathcal{O}(T_{in}n^{\log(n/\epsilon)})$ with error $\epsilon$.*

Note that here we actually perform the state $|t'_{m+1}\rangle = |g(1)\rangle|g(2)\rangle\cdots|g(m+1)\rangle|t_{m+1}\rangle$ from states $\{|g(1)\rangle|g(2)\rangle\cdots|g(m+1)\rangle|s_i\rangle\}_{i=1}^{m+1}$, and each state $(\prod_{j=1}^{m+1}|g(j)\rangle)|s_i\rangle$ could be performed by oracle $U_H$ on state $(\prod_{j=1}^{m+1}|g(j)\rangle)|0\rangle$. Denote $U_{m+1}$ as the unitary which generates the state $\left[\prod_{j=1}^{m+1}|g(j)\rangle\right]|t_{m+1}\rangle$ from the state $\left[\prod_{j=1}^{m+1}|g(j)\rangle\right]|0\rangle$. The operation in step (c) of procedure (11): $\prod_{m=1}^{l}[R_m \otimes |0\rangle\langle 0| + I \otimes |1\rangle\langle 1|]$ could be performed by $\prod_{m=1}^{l}\left\{\left[U_{m+1}^\dagger(I - 2|0\rangle\langle 0|)U_{m+1}\right] \otimes |0\rangle\langle 0| + I \otimes |1\rangle\langle 1|\right\}$.

## A.7   THE PROOF OF THEOREM 6

Denote $P_l$ as the probability of resulting $0$ after the measurement in Step 4 of Algorithm 3. In order to generate the required state, the measurement in Step 4 needs to be performed for $\mathcal{O}(1/P_l)$ times. The Lemma 4 provides a lower bound on $P_l$.

**Lemma 4.** *The probability of resulting $0$ after the measurement in Step 4 of Algorithm 3 as the lower bound $P_l \geq \frac{(r-l)\epsilon^2}{4\|H\|_F^2}$.*

*Proof.* Suppose $\lambda_1^2 \geq \lambda_2^2 \geq \cdots \geq \lambda_r^2$, where $\lambda_i$ is the eigenvalue of $H$. Since state $|t_m\rangle$ is the linear sum of $\{|h_j\rangle\}_{j=1}^{d}$, we can assume that $|t_m\rangle$ has the decomposition $|t_m\rangle = \sum_{i=1}^{r} w_{mi}|u_i\rangle$, for all

$m = 1, 2, \cdots, l$, where $\sum_{i=1}^{r} w_{mi} w_{ni} = \delta_{mn}$. There is:

$$
\begin{aligned}
P_l &= \frac{1}{\|\boldsymbol{H}\|_F^2} \sum_{j=1}^{d} \left[ \|\boldsymbol{h}_j\|^2 \|\boldsymbol{h}_j\rangle - \sum_{m=1}^{l} |\boldsymbol{t}_m\rangle\langle\boldsymbol{t}_m|\boldsymbol{h}_j\rangle\|^2 \right] \\
&= \frac{1}{\|\boldsymbol{H}\|_F^2} \sum_{j=1}^{d} \left[ \|\boldsymbol{h}_j\|^2 - \sum_{m=1}^{l} \|\boldsymbol{h}_j\|^2 |\langle\boldsymbol{t}_m|\boldsymbol{h}_j\rangle|^2 \right] \\
&= 1 - \frac{1}{\|\boldsymbol{H}\|_F^2} \sum_{j=1}^{d} \sum_{m=1}^{l} \left[ \sum_{i=1}^{r} w_{mi} \lambda_i u_i^{(j)} \right]^2 \\
&= 1 - \frac{1}{\|\boldsymbol{H}\|_F^2} \sum_{j=1}^{d} \sum_{m=1}^{l} \left[ \sum_{i=1}^{r} w_{mi}^2 \lambda_i^2 (u_i^{(j)})^2 + \sum_{i \neq k} w_{mi} w_{mk} \lambda_i \lambda_k u_i^{(j)} u_k^{(j)} \right] \\
&= 1 - \frac{1}{\|\boldsymbol{H}\|_F^2} \sum_{m=1}^{l} \sum_{i=1}^{r} w_{mi}^2 \lambda_i^2 \\
&= 1 - \frac{1}{\|\boldsymbol{H}\|_F^2} \sum_{i=1}^{r} c_i \lambda_i^2,
\end{aligned}
$$

where $c_i = \sum_{m=1}^{l} w_{mi}^2$.

Consider the $r$-dimensional vector $\boldsymbol{w}_m = \sum_{i=1}^{r} w_{mi} \boldsymbol{e}_i$. The vector set $\{\boldsymbol{w}_m\}_{m=1}^{l}$ forms an orthogonal basis in a $l$-dimensional subspace. Note that we can add $\boldsymbol{w}_{l+1}, \cdots \boldsymbol{w}_r$ such that $\{\boldsymbol{w}_m\}_{m=1}^{r}$ forms an orthonormal basis in the whole $r$-dimensional space. Denote matrix $W = (\boldsymbol{w}_1, \boldsymbol{w}_2, \cdots, \boldsymbol{w}_r)$. Since $W^T W = I$. Since $W$ is unitary, there is:

$$
\sum_{m=1}^{r} w_{mi}^2 = 1, \forall i \in [r]. \tag{20}
$$

Thus we have the upper bound: $c_i = \sum_{m=1}^{l} w_{mi}^2 \leq \sum_{m=1}^{r} w_{mi}^2 = 1$. Note that $\sum_{i=1}^{r} c_i = \sum_{i=1}^{r} \sum_{m=1}^{l} w_{mi}^2 = \sum_{m=1}^{l} \sum_{i=1}^{r} w_{mi}^2 = l$, so there is:

$$
P_l \geq 1 - \frac{1}{\|\boldsymbol{H}\|_F^2} \sum_{i=1}^{l} \lambda_i^2 = \frac{\sum_{i=l+1}^{r} \lambda_i^2}{\|\boldsymbol{H}\|_F^2}. \tag{21}
$$

Note that for the case $|\lambda_i| \leq \epsilon/2$, $\tilde{\lambda}_i = 0$ is a good estimation for the NCF problem, so we could further assume $|\lambda_i| > \epsilon/2$ for the general case and bound the inequality (21) as $P_l \geq \frac{(r-l)\epsilon^2}{4\|\boldsymbol{H}\|_F^2}$.

$\square$

The error of implementing $|\boldsymbol{t}_{m+1}\rangle$ comes from the imperfect implementing of the linear-combination-of-states operation and the error of calculating $\boldsymbol{x}$. The former has been analyzed in Theorem 9. The error of calculating $\boldsymbol{x}$ is more complex.

Define vector $\boldsymbol{y} = -Z_{m+1}\boldsymbol{x}$. Note that all parameters $c_{ij} = \langle\boldsymbol{s}_i|\boldsymbol{s}_j\rangle, i, j \in [m]$ and $b_j = \langle\boldsymbol{s}_{m+1}|\boldsymbol{s}_j\rangle, j \in [m]$ are estimated by Hadamard test, which takes time $\mathcal{O}(T_H\epsilon'^{-2})$ with error bounded in $\epsilon'$ (see Appendix A.10 for more information). Thus the calculation on vector $\boldsymbol{y} = C_m^{-1}\boldsymbol{b}$ would have an error. Define matrix $\tilde{\boldsymbol{C}}_m = \boldsymbol{C}_m + \Delta\boldsymbol{C}_m$ and $\tilde{\boldsymbol{b}} = \boldsymbol{b} + \Delta\boldsymbol{b}$ which are estimations on $\boldsymbol{C}_m$ and $\boldsymbol{b}$. Suppose $|\tilde{c}_{ij} - c_{ij}| \leq \epsilon_1$ and $|\tilde{b}_i - b_i| \leq \epsilon_1$ are error bounds for $c_{ij}$ and $b_i$, respectively, $\forall i, j \in [m]$. Denote $\tilde{\boldsymbol{y}} = \tilde{\boldsymbol{C}}_m^{-1}\boldsymbol{b}$ as the solution to the approximate linear equation and $\Delta\boldsymbol{y} = \tilde{\boldsymbol{y}} - \boldsymbol{y}$ as the error to $\boldsymbol{y}$. We have:

$$
\begin{cases}
\boldsymbol{C}_m\boldsymbol{y} = \boldsymbol{b}, \\
(\boldsymbol{C}_m + \Delta\boldsymbol{C}_m)(\boldsymbol{y} + \Delta\boldsymbol{y}) = (\boldsymbol{b} + \Delta\boldsymbol{b}).
\end{cases}
$$

So there is:

$$
\begin{aligned}
\|\Delta \boldsymbol{y}\| &= \|(\boldsymbol{C}_m + \Delta \boldsymbol{C}_m)^{-1}(\Delta \boldsymbol{b} - \Delta \boldsymbol{C}_m \cdot \boldsymbol{C}_m^{-1}\boldsymbol{b})\| \\
&\leq \|\boldsymbol{C}_m^{-1}\| \cdot \|(\boldsymbol{I} + \boldsymbol{C}_m^{-1}\Delta \boldsymbol{C}_m)^{-1}\| \cdot (\|\Delta \boldsymbol{b}\| + \|\Delta \boldsymbol{C}_m \cdot \boldsymbol{C}_m^{-1}\boldsymbol{b}\|) \\
&\leq \|\boldsymbol{C}_m^{-1}\| \cdot \frac{1}{1 - \|\boldsymbol{C}_m^{-1}\Delta \boldsymbol{C}_m\|} \cdot (\|\Delta \boldsymbol{b}\| + \|\Delta \boldsymbol{C}_m\|\|\boldsymbol{C}_m^{-1}\|\|\boldsymbol{b}\|) \\
&\leq \frac{\|\boldsymbol{C}_m^{-1}\|}{1 - \|\boldsymbol{C}_m^{-1}\|m\epsilon_1} \cdot (\sqrt{m}\epsilon_1 + m^{3/2}\epsilon_1\|\boldsymbol{C}_m^{-1}\|) \leq \frac{2m^{3/2}\|\boldsymbol{C}_m^{-1}\|^2\epsilon_1}{1 - \|\boldsymbol{C}_m^{-1}\|m\epsilon_1}.
\end{aligned}
$$

The norm $\|\boldsymbol{C}_m^{-1}\|$ here denotes the largest norm of eigenvalues of matrix $\boldsymbol{C}_m^{-1}$. Thus, for $\epsilon_2 = 3m^{3/2}\|\boldsymbol{C}_m^{-1}\|^2\epsilon_1$, we have bound $\|\Delta \boldsymbol{y}\| \leq \epsilon_2$.

For $|\boldsymbol{t}_{m+1}\rangle = \sum_{i=1}^{m+1} x_{m+1,i}|\boldsymbol{s}_i\rangle$ and $|\tilde{\boldsymbol{t}}_{m+1}\rangle = \sum_{i=1}^{m+1} \tilde{x}_{m+1,i}|\boldsymbol{s}_i\rangle$, there is:

$$
\begin{aligned}
\langle \boldsymbol{t}_{m+1}|\tilde{\boldsymbol{t}}_{m+1}\rangle &= \sum_{i=1}^{m+1}\sum_{j=1}^{m+1} x_{m+1,i}c_{ij}\tilde{x}_{m+1,j} = \frac{1}{Z_{m+1}\tilde{Z}_{m+1}}[\sum_{i=1,j=1}^{m,m} y_i c_{ij}\tilde{y}_j - \sum_{i=1}^{m} y_i b_i - \sum_{j=1}^{m} b_j \tilde{y}_j + 1] \\
&= \frac{1}{Z_{m+1}\tilde{Z}_{m+1}}[1 - \sum_{i=1}^{m} y_i b_i] = \frac{Z_{m+1}}{\tilde{Z}_{m+1}}.
\end{aligned}
$$

Note that the form $|\tilde{\boldsymbol{t}}_{m+1}\rangle = \sum_{i=1}^{m+1} \tilde{x}_{m+1,i}|\boldsymbol{s}_i\rangle$ is not a normalized state:

$$
\begin{aligned}
\||\tilde{\boldsymbol{t}}_{m+1}\rangle\|^2 &= \sum_{i=1}^{m+1}\sum_{j=1}^{m+1} \tilde{x}_{m+1,i}c_{ij}\tilde{x}_{m+1,j} = \frac{1}{\tilde{Z}_{m+1}^2}[\sum_{i=1,j=1}^{m,m} \tilde{y}_i c_{ij}\tilde{y}_j + 1 - 2\sum_{i=1}^{m} \tilde{y}_i b_i] \\
&= \frac{1}{\tilde{Z}_{m+1}^2}[\sum_{i=1,j=1}^{m,m} \Delta y_i c_{ij}\Delta y_j + 2\sum_{i=1,j=1}^{m,m} \Delta y_i c_{ij} y_j + \sum_{i=1,j=1}^{m,m} y_i c_{ij} y_j + 1 - 2\sum_{i=1}^{m} \Delta y_i b_i - 2\sum_{i=1}^{m} y_i b_i] \\
&= \frac{1}{\tilde{Z}_{m+1}^2}[Z_{m+1}^2 + \sum_{i=1,j=1}^{m,m} \Delta y_i c_{ij}\Delta y_j].
\end{aligned}
$$

Thus, the overlap between state $|\boldsymbol{t}_{m+1}\rangle$ and $|\tilde{\boldsymbol{t}}_{m+1}\rangle$ is:

$$
\frac{\langle \boldsymbol{t}_{m+1}|\tilde{\boldsymbol{t}}_{m+1}\rangle}{\||\tilde{\boldsymbol{t}}_{m+1}\rangle\|} = \frac{Z_{m+1}}{\sqrt{Z_{m+1}^2 + \Delta \boldsymbol{y}^T \boldsymbol{C}_m \Delta \boldsymbol{y}}}. \tag{22}
$$

There is:

$$
\||\boldsymbol{t}_{m+1}\rangle - |\tilde{\boldsymbol{t}}_{m+1}\rangle/\||\tilde{\boldsymbol{t}}_{m+1}\rangle\|\| = \sqrt{2 - 2\frac{Z_{m+1}}{\sqrt{Z_{m+1}^2 + \Delta \boldsymbol{y}^T \boldsymbol{C}_m \Delta \boldsymbol{y}}}} \tag{23}
$$

$$
\leq \sqrt{2 - 2\frac{Z_{m+1}}{\sqrt{Z_{m+1}^2 + \|\Delta \boldsymbol{y}\|^2\|\boldsymbol{C}_m\|}}} \tag{24}
$$

$$
\leq \sqrt{2 - 2\frac{Z_{m+1}}{Z_{m+1} + \frac{\|\boldsymbol{C}_m\|\|\Delta \boldsymbol{y}\|^2}{2Z_{m+1}}}} \tag{25}
$$

$$
\leq \frac{\|\boldsymbol{C}_m\|^{1/2}\|\Delta \boldsymbol{y}\|}{Z_{m+1}} \tag{26}
$$

$$
\leq \frac{m^{1/2}\epsilon_2}{Z_{m+1}} = \frac{3m^2\|\boldsymbol{C}_m^{-1}\|^2\epsilon_1}{Z_{m+1}}. \tag{27}
$$

Let $\epsilon_3/2 = \frac{3m^2\|\boldsymbol{C}_m^{-1}\|^2\epsilon_1}{Z_{m+1}}$. It is clear that to obtain the linear combination form $|\tilde{\boldsymbol{t}}_{m+1}\rangle = \sum_{i=1}^{m+1} \tilde{x}_{m+1,i}|\boldsymbol{s}_i\rangle$ takes time $\mathcal{O}(m^3 + m^2 T_H \epsilon_1^{-2})$, where $\||\tilde{\boldsymbol{t}}_{m+1}\rangle/\||\tilde{\boldsymbol{t}}_{m+1}\rangle\| - |\boldsymbol{t}_{m+1}\rangle\| \leq$

$\epsilon_3/2$. By Theorem 9, the implementation of state $|\tilde{t}'_{m+1}\rangle = [\prod_{j=1}^{m+1}|g(j)\rangle]|\tilde{t}_{m+1}\rangle$ takes time $\mathcal{O}(T_H(m+1)^{\log(2(m+1)/\epsilon_3)})$ with error bounds $\epsilon_3/2$. Thus, we can implement state $|t'_{m+1}\rangle = [\prod_{j=1}^{m+1}|g(j)\rangle]|t_{m+1}\rangle$ by unitary in time $\mathcal{O}(m^3 + m^2 T_H \epsilon_1^{-2} + T_H(m+1)^{\log(2(m+1)/\epsilon_3)})$ with error bounds in $\epsilon_3$.

Now we consider the influence of imperfect implementation of state $|\tilde{t}'_m\rangle$ to Algorithm 3. For simplicity, we neglect the term $[\prod_{j=1}^{m}|g(j)\rangle]$ in state $|t'_m\rangle = [\prod_{j=1}^{m}|g(j)\rangle]|t_m\rangle$. Denote $\Pi_l = \prod_{i=1}^{l}(I - 2|t_i\rangle\langle t_i|)$ and $\tilde{\Pi}_l = \prod_{i=1}^{l}(I - 2|\tilde{t}_i\rangle\langle\tilde{t}_i|)$. There is $\|\Pi_l - \tilde{\Pi}_l\| \leq 2l\epsilon_3$.

Note that the state

$$|\phi_1^{(l)}\rangle = \frac{1}{\|\boldsymbol{H}\|_F}\sum_{j=1}^{d}\|\boldsymbol{h}_j\|\,|j\rangle\left\{\left[|\boldsymbol{h}_j\rangle - \sum_{m=1}^{l}|\boldsymbol{t}_m\rangle\langle\boldsymbol{t}_m|\boldsymbol{h}_j\rangle\right]|0\rangle - \sum_{m=1}^{l}|\boldsymbol{t}_m\rangle\langle\boldsymbol{t}_m|\boldsymbol{h}_j\rangle|1\rangle\right\}$$

in step 3 of Algorithm 3 can be written as:

$$\frac{1}{\|\boldsymbol{H}\|_F}\sum_{j=1}^{d}\|\boldsymbol{h}_j\|\,|j\rangle\left[\frac{\Pi_l + I}{2}|\boldsymbol{h}_j\rangle|0\rangle + \frac{\Pi_l - I}{2}|\boldsymbol{h}_j\rangle|1\rangle\right].$$

The probability of generating 0 after the measurement on the last register is:

$$P_l = \frac{1}{\|\boldsymbol{H}\|_F^2}\sum_{j=1}^{d}\|\boldsymbol{h}_j\|^2\|\frac{\Pi_l + I}{2}|\boldsymbol{h}_j\rangle\|^2.$$

Similarly we define $|\tilde{\phi}_1^{(l)}\rangle = \frac{1}{\|\boldsymbol{H}\|_F}\sum_{j=1}^{d}\|\boldsymbol{h}_j\|\,|j\rangle[\frac{\tilde{\Pi}_l + I}{2}|\boldsymbol{h}_j\rangle|0\rangle + \frac{\tilde{\Pi}_l - I}{2}|\boldsymbol{h}_j\rangle|1\rangle]$,

and $\tilde{P}_l = \frac{1}{\|\boldsymbol{H}\|_F}\sum_{j=1}^{d}\|\boldsymbol{h}_j\|^2\|\frac{\tilde{\Pi}_l + I}{2}|\boldsymbol{h}_j\rangle\|^2$ for the approximate case.

Since the objective of step 5 is to obtain the index $g(l+1)$ such that the column state $|\boldsymbol{h}_{g(l+1)}\rangle = |\boldsymbol{s}_{l+1}\rangle$ is linearly independent from basis $\{|\boldsymbol{s}_i\rangle\}_{i=1}^{l}$, we define $P_l^{false}$ as the probability of selecting out the state $|\boldsymbol{s}_{l+1}\rangle \in \{|\boldsymbol{s}_i\rangle\}_{i=1}^{l}$ in the approximate case. Note that for state $|\boldsymbol{h}_j\rangle \in \{|\boldsymbol{s}_i\rangle\}_{i=1}^{l}$, $(\Pi_l + I)|\boldsymbol{h}_j\rangle = 0$, so there is:

$$P_l^{false} = \frac{1}{\tilde{P}_l}\frac{1}{\|\boldsymbol{H}\|_F^2}\sum_{j:|\boldsymbol{h}_j\rangle\in\{|\boldsymbol{s}_i\rangle\}_{i=1}^{l}}\|\boldsymbol{h}_j\|^2\|\frac{\tilde{\Pi}_l + I}{2}|\boldsymbol{h}_j\rangle\|^2$$

$$= \frac{\sum_{j:|\boldsymbol{h}_j\rangle\in\{|\boldsymbol{s}_i\rangle\}_{i=1}^{l}}\|\boldsymbol{h}_j\|^2\|\frac{\tilde{\Pi}_l + I}{2}|\boldsymbol{h}_j\rangle\|^2}{\sum_{j=1}^{d}\|\boldsymbol{h}_j\|^2\|\frac{\tilde{\Pi}_l + I}{2}|\boldsymbol{h}_j\rangle\|^2}$$

$$= \frac{\sum_{j:|\boldsymbol{h}_j\rangle\in\{|\boldsymbol{s}_i\rangle\}_{i=1}^{l}}\|\boldsymbol{h}_j\|^2\|\frac{\tilde{\Pi}_l - \Pi_l}{2}|\boldsymbol{h}_j\rangle\|^2}{\sum_{j=1}^{d}\|\boldsymbol{h}_j\|^2(1/2 + \langle\boldsymbol{h}_j|\tilde{\Pi}_l - \Pi_l|\boldsymbol{h}_j\rangle/2 + \langle\boldsymbol{h}_j|\Pi_l|\boldsymbol{h}_j\rangle/2)}$$

$$\leq \frac{l^2\epsilon_3^2}{P_l - l\epsilon_3} \leq \frac{l^2\epsilon_3^2}{\frac{(r-l)\epsilon^2}{4\|\boldsymbol{H}\|_F^2} - l\epsilon_3}.$$

Let $\epsilon_3 = \frac{\epsilon^2}{8(r-1)\|\boldsymbol{H}\|_F^2}$, there is:

$$\sum_{l=0}^{r-1}P_l^{false} \leq \sum_{l=0}^{r-1}\frac{l^2\epsilon_3^2}{2(r-1)(r-l)\epsilon_3 - l\epsilon_3} \leq \sum_{l=0}^{r-1}l\epsilon_3 = \frac{r(r-1)}{2}\frac{\epsilon^2}{8(r-1)\|\boldsymbol{H}\|_F^2} = \frac{r\epsilon^2}{16\|\boldsymbol{H}\|_F^2} \leq \frac{1}{4}. \tag{28}$$

Thus, by choosing $\epsilon_3 = \frac{\epsilon^2}{8(r-1)\|\boldsymbol{H}\|_F^2}$, Algorithm 3 could select out a complete basis $\{|\boldsymbol{s}_i\rangle\}_{i=1}^{r}$ with probability at least $\frac{3}{4}$.

Note that in Algorithm 3 we need to perform operations $R_i = I - 2|\boldsymbol{t}_i'\rangle\langle\boldsymbol{t}_i'|$ for $i = 1, 2, \cdots, r-1$, which needs the information of parameters $\{c_{ij}\}_{i=1,j=1}^{r-1,r-1}$. In order to guarantee the success probability of Algorithm 3 (Equation (28)), the estimation on each $c_{ij} = \langle\boldsymbol{s}_i|\boldsymbol{s}_j\rangle$ should have error bound $\epsilon_1 = \min_{m\in[r-2]} \frac{Z_{m+1}}{6m^2\|\boldsymbol{C}_m^{-1}\|^2}\epsilon_3 \geq \frac{\min_{m\in[r-1]} Z_m}{48r^3\|\boldsymbol{C}_r^{-1}\|^2\|\boldsymbol{H}\|_F^2}\epsilon^2$. Thus, the estimation on each $c_{ij}$ takes time $\mathcal{O}(T_H r^6\|\boldsymbol{H}\|_F^4\epsilon^{-4})$ and the estimation on the parameter group $\{c_{ij}\}_{i=1,j=1}^{r-1,r-1}$ takes time $\mathcal{O}(T_H r^8\|\boldsymbol{H}\|_F^4\epsilon^{-4})$. In order to obtain the description $|\boldsymbol{t}_m\rangle = \sum_{i=1}^{m} x_{m,i}|\boldsymbol{s}_i\rangle$, additional time is required for solving (m-1)-dimensional equations for $m = 2, 3, \cdots r-1$, which results the time complexity $\mathcal{O}(r^4)$ in total. With given parameters $\{x_{m,i}\}_{i=1}^{m}$, the implementation of operation $R_m = I - 2|\boldsymbol{t}_m'\rangle\langle\boldsymbol{t}_m'|$ takes time $\mathcal{O}(T_H(m+1)^{\log(2(m+1)/\epsilon_3)}) \leq \mathcal{O}(T_H \cdot r^{2\log(4r\|\boldsymbol{H}\|_F/\epsilon)})$.

Denote $T_{basis}$ as the required time to implement Algorithm 3 and $T_{R_i}$ as the required time to implement operation $R_i$. Since in each iteration of $l \in [r-1]$, Algorithm 3 refers operation $R_1, R_2, \cdots, R_l$ for $1/P_l$ times, there is:

$$\mathcal{O}(T_{basis}) = \mathcal{O}(T_H r^8\|\boldsymbol{H}\|_F^4\epsilon^{-4}) + \sum_{l=0}^{r-1}\frac{1}{P_l}\sum_{m=1}^{l}\mathcal{O}(T_{R_i})$$

$$\leq \mathcal{O}(T_H r^8\|\boldsymbol{H}\|_F^4\epsilon^{-4}) + \frac{4\|\boldsymbol{H}\|_F^2}{\epsilon^2}\mathcal{O}(T_H r^{2\log(4r\|\boldsymbol{H}\|_F/\epsilon)})\sum_{l=0}^{r-1}\frac{l}{r-l}$$

$$= \mathcal{O}(T_H\|\boldsymbol{H}\|_F^2\epsilon^{-2}(r^8\|\boldsymbol{H}\|_F^2\epsilon^{-2} + r^{1+2\log(4r\|\boldsymbol{H}\|_F/\epsilon)}))$$

$$\leq \mathcal{O}(T_H\text{poly}(r)\epsilon^{-2}r^{2\log(4r\|\boldsymbol{H}\|_F/\epsilon)}).$$

## A.8 THE PROOF OF LEMMA 2

*Proof.* Define the index function $g : [r] \to [d]$ such that $\boldsymbol{s}_i = \boldsymbol{h}_{g(i)}, \forall i \in [r]$. Consider the eigen-decomposition of matrix $\boldsymbol{H}$:

$$\boldsymbol{H} = \sum_{j=1}^{r}\lambda_j\boldsymbol{u}_j\boldsymbol{u}_j^T. \tag{29}$$

It is natural to generate the decomposition:

$$\boldsymbol{h}_j = \sum_{i=1}^{r}\lambda_i\boldsymbol{u}_i u_i^{(j)}, \tag{30}$$

$$h_{jk} = \sum_{i=1}^{r}\lambda_i u_i^{(j)}u_i^{(k)}. \tag{31}$$

Define the $r \times r$ dimensional matrix $\boldsymbol{C} = (\boldsymbol{h}_{g(1)}^T, \boldsymbol{h}_{g(2)}^T, \cdots, \boldsymbol{h}_{g(r)}^T)^T(\boldsymbol{h}_{g(1)}, \boldsymbol{h}_{g(2)}, \cdots, \boldsymbol{h}_{g(r)})$. There is:

$$\{\boldsymbol{h}_{g(i)}\}_{i=1}^{r} \text{ is linear independent} \Leftrightarrow det(\boldsymbol{C}) \neq 0. \tag{32}$$

Denote the $jk$-th element of $\boldsymbol{C}$ as $c_{jk}$. Since $c_{jk} = \boldsymbol{h}_j^T\boldsymbol{h}_k = \sum_{i=1}^{r}\lambda_i^2 u_i^{(j)}u_i^{(k)}$, there is:

$$det(\boldsymbol{C}) = \begin{vmatrix} \sum_{i=1}^{r}\lambda_i^2 u_i^{(g(1))}u_i^{(g(1))} & \cdots & \sum_{i=1}^{r}\lambda_i^2 u_i^{(g(1))}u_i^{(g(r))} \\ \vdots & \ddots & \vdots \\ \sum_{i=1}^{r}\lambda_i^2 u_i^{(g(r))}u_i^{(g(1))} & \cdots & \sum_{i=1}^{r}\lambda_i^2 u_i^{(g(r))}u_i^{(g(r))} \end{vmatrix} \tag{33}$$

$$= \sum_{i_1=1}^{r}\sum_{i_2=1}^{r}\cdots\sum_{i_r=1}^{r} \begin{vmatrix} \lambda_{i_1}^2 u_{i_1}^{(g(1))}u_{i_1}^{(g(1))} & \cdots & \lambda_{i_r}^2 u_{i_r}^{(g(1))}u_{i_r}^{(g(r))} \\ \vdots & \ddots & \vdots \\ \lambda_{i_1}^2 u_{i_1}^{(g(r))}u_{i_1}^{(g(1))} & \cdots & \lambda_{i_r}^2 u_{i_r}^{(g(r))}u_{i_r}^{(g(r))} \end{vmatrix} \tag{34}$$

$$= \sum_{i_1=1}^{r}\sum_{i_2=1}^{r}\cdots\sum_{i_r=1}^{r}(\prod_{j=1}^{r}\lambda_{i_j}^2)(\prod_{j=1}^{r}u_{i_j}^{(g(j))}) \begin{vmatrix} u_{i_1}^{(g(1))} & \cdots & u_{i_r}^{(g(1))} \\ \vdots & \ddots & \vdots \\ u_{i_1}^{(g(r))} & \cdots & u_{i_r}^{(g(r))} \end{vmatrix}. \tag{35}$$

On the other hand, construct the matrix $\boldsymbol{H}'$ whose $jk$-th element is $h'_{jk} = h_{g(j),g(k)}$. There is:

$$
det(\boldsymbol{H}') = \begin{vmatrix} \sum_{i=1}^{r} \lambda_i u_i^{(g(1))} u_i^{(g(1))} & \cdots & \sum_{i=1}^{r} \lambda_i u_i^{(g(1))} u_i^{(g(r))} \\ \vdots & \ddots & \vdots \\ \sum_{i=1}^{r} \lambda_i u_i^{(g(r))} u_i^{(g(1))} & \cdots & \sum_{i=1}^{r} \lambda_i u_i^{(g(r))} u_i^{(g(r))} \end{vmatrix} \tag{36}
$$

$$
= \sum_{i_1=1}^{r} \sum_{i_2=1}^{r} \cdots \sum_{i_r=1}^{r} \begin{vmatrix} \lambda_{i_1} u_{i_1}^{(g(1))} u_{i_1}^{(g(1))} & \cdots & \lambda_{i_r} u_{i_r}^{(g(1))} u_{i_r}^{(g(r))} \\ \vdots & \ddots & \vdots \\ \lambda_{i_1} u_{i_1}^{(g(r))} u_{i_1}^{(g(1))} & \cdots & \lambda_{i_r} u_{i_r}^{(g(r))} u_{i_r}^{(g(r))} \end{vmatrix} \tag{37}
$$

$$
= \sum_{i_1=1}^{r} \sum_{i_2=1}^{r} \cdots \sum_{i_r=1}^{r} (\prod_{j=1}^{r} \lambda_{i_j})(\prod_{j=1}^{r} u_{i_j}^{(g(j))}) \begin{vmatrix} u_{i_1}^{(g(1))} & \cdots & u_{i_r}^{(g(1))} \\ \vdots & \ddots & \vdots \\ u_{i_1}^{(g(r))} & \cdots & u_{i_r}^{(g(r))} \end{vmatrix} . \tag{38}
$$

Note that the determinant in eq(35) and eq(38) is non-zero only if $i_m \neq i_n$ for any different $m, n \in [r]$. Consider the summation of $i_j$ for all $j \in [r]$ over $\{1, 2, \cdots, r\}$, there is:

$$
det(\boldsymbol{C})/\prod_{i=1}^{r} \lambda_i^2 = det(\boldsymbol{H}')/\prod_{i=1}^{r} \lambda_i \tag{39}
$$

Thus the problem about whether group $\{\boldsymbol{h}_{g(i)}\}_{i=1}^{r}$ is linearly independent could be solved by calculating the determinant of matrix $\boldsymbol{H}'$. Since $\boldsymbol{H}'$ is a $r \times r$ dimensional matrix, $det(\boldsymbol{H}')$ could be calculated in $\mathcal{O}(r^3)$ time (Schwarzenberg-Czerny, 1995). We could claim that the group $\{\boldsymbol{h}_{g(i)}\}_{i=1}^{r}$ is linearly independent if $det(\boldsymbol{H}') \neq 0$, or $\{\boldsymbol{h}_{g(i)}\}_{i=1}^{r}$ is linear dependent if $det(\boldsymbol{H}') = 0$. $\square$

### A.9 THE PROOF OF THEOREM 7

*Proof.* For $|\Delta c_{ij}| \leq \epsilon_1$ and $|\Delta b_j| \leq \epsilon_2$, there is:

$$
\|\Delta \boldsymbol{C}\| \leq r\epsilon_1 \text{ and } \|\Delta \boldsymbol{b}\| \leq \sqrt{r}\epsilon_2.
$$

The matrix norm $\| \cdot \|$ here denotes the largest singular value of the matrix. Note that elements of matrix $\boldsymbol{C}$ and vector $\boldsymbol{b}$ are overlap of quantum states, which are bounded in $[-1, 1]$, so there is:

$$
\|\boldsymbol{C}\| \leq r \text{ and } \|\boldsymbol{b}\| \leq \sqrt{r}.
$$

We have:

$$
\begin{aligned}
\|\Delta \boldsymbol{x}\| &= \|(\boldsymbol{C} + \Delta \boldsymbol{C})^{-1}(\Delta \boldsymbol{b} - \Delta \boldsymbol{C} \cdot \boldsymbol{C}^{-1}\boldsymbol{b})\| \\
&\leq \|\boldsymbol{C}^{-1}\| \cdot \|(I + \boldsymbol{C}^{-1}\Delta \boldsymbol{C})^{-1}\| \cdot (\|\Delta \boldsymbol{b}\| + \|\Delta \boldsymbol{C} \cdot \boldsymbol{C}^{-1}\boldsymbol{b}\|) \\
&\leq \|\boldsymbol{C}^{-1}\| \cdot \frac{1}{1 - \|\boldsymbol{C}^{-1}\Delta \boldsymbol{C}\|} \cdot (\|\Delta \boldsymbol{b}\| + \|\Delta \boldsymbol{C}\|\|\boldsymbol{C}^{-1}\|\|\boldsymbol{b}\|) \\
&\leq \frac{\|\boldsymbol{C}^{-1}\|}{1 - \|\boldsymbol{C}^{-1}\|r\epsilon_1} \cdot (\sqrt{r}\epsilon_2 + r^{3/2}\epsilon_1\|\boldsymbol{C}^{-1}\|) \leq \frac{\epsilon}{2\sqrt{r}}.
\end{aligned}
$$

Thus, for $\boldsymbol{u}_t = \sum_{j=1}^{r} x_j \boldsymbol{s}_j$ and $\tilde{\boldsymbol{u}}_t = \sum_{j=1}^{r} \tilde{x}_j \boldsymbol{s}_j$, there is:

$$
\|\boldsymbol{u}_t - \tilde{\boldsymbol{u}}_t\| = \sqrt{\Delta \boldsymbol{x}^T \boldsymbol{C} \Delta \boldsymbol{x}} \leq \|\Delta \boldsymbol{x}\| \cdot \|\boldsymbol{C}\|^{1/2} \leq \frac{\epsilon}{2\sqrt{r}} \cdot \sqrt{r} = \frac{\epsilon}{2}.
$$

$\square$

### A.10 THE ESTIMATION OF $c_{ij} = \langle s_i | s_j \rangle$

The overlap $c_{ij} = \langle \boldsymbol{s}_i | \boldsymbol{s}_j \rangle = \langle \boldsymbol{h}_{g(i)} | \boldsymbol{h}_{g(j)} \rangle$ can be estimated by the Hadamard Test. We provide the detail in Algorithm 6:

**Proposition 4.** *Algorithm 6 present the $\epsilon$-estimation to the overlap $c_{ij} = \langle \boldsymbol{s}_i | \boldsymbol{s}_j \rangle$ with probability at least $1 - \delta$ with running time $\mathcal{O}(T_H \epsilon^{-2} \log(1/\delta))$.*

---

**Algorithm 6** $c_{ij}$ estimation

---

**Input:** Quantum access to oracle $U_H$. The index number $i$ and $j$. The precision parameter $\epsilon$. The probability error bound $\delta$.

**Output:** An estimation $\tilde{c}_{ij}$ to the value $c_{ij} = \langle s_i | s_j \rangle$, such that $\tilde{c}_{ij} \in [c_{ij} - \epsilon \, , \, c_{ij} + \epsilon]$ with probability at least $1 - \delta$.

1: **for** $k = 1$ to $n = \lceil \frac{2}{\epsilon^2} \log(\frac{2}{\delta}) \rceil + 1$ **do**
2:      Create state $[|s_i\rangle|0\rangle + |s_j\rangle|1\rangle]/\sqrt{2}$.
3:      Apply the Hadmard gate on the second register to obtain the state $\frac{|s_i\rangle + |s_j\rangle}{2}|0\rangle + \frac{|s_i\rangle - |s_j\rangle}{2}|1\rangle$.
4:      Measure the second register and record the result.
5: **end for**
6: Count the number of resulting 0 in step 4 as $m$. Output $2m/n - 1$ as the estimation to $c_{ij}$.

---

State $\frac{|s_i\rangle|0\rangle + |s_j\rangle|1\rangle}{\sqrt{2}} = \frac{|h_{g(i)}\rangle|0\rangle + |h_{g(j)}\rangle|1\rangle}{\sqrt{2}}$ in step 2 could be generated by performing the following procedure on state $|g(i)\rangle|g(j)\rangle|0\rangle|0\rangle$:

$$|g(i)\rangle|g(j)\rangle|0\rangle|0\rangle \xrightarrow{H} |g(i)\rangle|g(j)\rangle|0\rangle \frac{|0\rangle + |1\rangle}{\sqrt{2}} \tag{40}$$

$$\xrightarrow{U_H \otimes |0\rangle\langle 0|} |g(i)\rangle|g(j)\rangle \frac{|h_{g(i)}\rangle|0\rangle + |0\rangle|1\rangle}{\sqrt{2}} \tag{41}$$

$$\xrightarrow{U_H \otimes |1\rangle\langle 1|} |g(i)\rangle|g(j)\rangle \frac{|h_{g(i)}\rangle|0\rangle + |h_{g(j)}\rangle|1\rangle}{\sqrt{2}} \tag{42}$$

$$\xrightarrow{\text{trace out the first two registers}} \frac{|h_{g(i)}\rangle|0\rangle + |h_{g(j)}\rangle|1\rangle}{\sqrt{2}}. \tag{43}$$

The Hadmard gate in (40) acts on the 4-th register. The gate $U_H \otimes |0\rangle\langle 0|$ in (41) acts on the 1-st, 3-rd and 4-th registers. The gate $U_H \otimes |1\rangle\langle 1|$ in (42) acts on the 2-nd, 3-rd and 4-th registers.

## A.11    THE ESTIMATION OF $b_i = \langle u_t | s_i \rangle$

The estimation to the overlap $b_i = \langle u_t | s_i \rangle$ is more complicated. Technics like Algorithm 6 is infeasible, due to the post-selection method for generating target state $|u_t\rangle$. Here we introduce another standard quantum algorithm named as Quantum Swap Test (Buhrman et al., 2001), which could estimate the square overlap between two quantum states $|\phi\rangle$ and $|\psi\rangle$. The circuit of the Quantum Swap Test is illustrated in Figure 1.

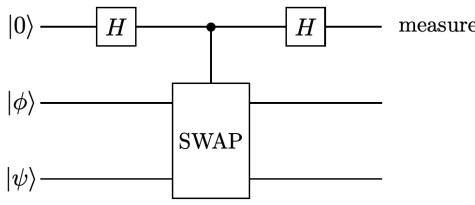

Figure 1: Circuit of the Quantum Swap Test

As shown in Figure 1, Quantum Swap Test performs the operation:

$$|0\rangle|\phi\rangle|\psi\rangle \rightarrow (H \otimes I)(|0\rangle\langle 0| \otimes I + |1\rangle\langle 1| \otimes U_{SWAP})(H \otimes I)|0\rangle|\phi\rangle|\psi\rangle. \tag{44}$$

The final state could be written as:

$$\frac{1}{2}|0\rangle(|\phi\rangle|\psi\rangle + |\psi\rangle|\phi\rangle) + \frac{1}{2}|1\rangle(|\phi\rangle|\psi\rangle - |\psi\rangle|\phi\rangle). \tag{45}$$

The $U_{SWAP}$ gate could be implemented in time $\mathcal{O}(\text{polylog}(d))$, which performs the swap transformation $|\phi\rangle|\psi\rangle \rightarrow |\psi\rangle|\phi\rangle$ for $d$-dimensional state $|\phi\rangle$ and $|\psi\rangle$. The measurement on the first qubit

produces 0 with probability $P_0 = \frac{1}{2}(1 + |\langle \phi | \psi \rangle|^2)$. Thus, by replacing step 2-3 in Algortihm 6 with the Quantum Swap Test operation, we could build an algorithm to estimate the square of the state overlap.

**Proposition 5.** *There exists a quantum algorithm which could present $\epsilon$-estimation to value $b_i^2 = |\langle \boldsymbol{u}_t | \boldsymbol{s}_i \rangle|^2$ with probability at least $1 - \delta$ in running time $\mathcal{O}(T_{Input} \text{polylog}(d)\epsilon^{-2} \log(1/\delta))$, where $T_{Input}$ is the time complexity to generate states $|\boldsymbol{u}_t\rangle$ and $|\boldsymbol{s}_i\rangle$.*

In order to estimate values $b_i = \langle \boldsymbol{u}_t | \boldsymbol{s}_i \rangle$ for $i \in [r]$, we need to discriminate the positive and negative of $b_i$. Note that for state $|\boldsymbol{u}_t\rangle$, the state $|-\boldsymbol{u}_t\rangle$ is also a target state which shares the same eigenvalue. So both states $|\boldsymbol{u}_t\rangle$ and $|-\boldsymbol{u}_t\rangle$ are legal outputs and indistinguishable for our algorithm in Section 3. Thus we analysis the value $b_i = sgn(u_t^{(k)})\langle \boldsymbol{u}_t | \boldsymbol{s}_i \rangle$ as the overlap between states $|\boldsymbol{u}_t\rangle$ and $|\boldsymbol{s}_i\rangle$, where $u_t^{(k)}$ is the $k$-th component of vector $\boldsymbol{u}_t$. Generally $k$ could be any index such that the corresponding component is non-zero.

Note that $b_i = sgn(u_t^{(k)})\langle \boldsymbol{u}_t | \boldsymbol{s}_i \rangle = -sgn(\langle \boldsymbol{u}_t | \boldsymbol{h}_k \rangle)\langle \boldsymbol{u}_t | \boldsymbol{s}_i \rangle$. Define two states $|\psi_+\rangle = \frac{1}{Z_+}(|\boldsymbol{h}_k\rangle + |\boldsymbol{h}_{g(i)}\rangle)$ and $|\psi_-\rangle = \frac{1}{Z_-}(|\boldsymbol{h}_k\rangle - |\boldsymbol{h}_{g(i)}\rangle)$, where $Z_\pm$ are normalized constants such that $Z_\pm^2 = 2 \pm 2\langle \boldsymbol{h}_k | \boldsymbol{h}_{g(i)} \rangle$. Then there is:

$$|\langle \boldsymbol{u}_t | \psi_+ \rangle|^2 = \frac{1}{Z_+^2}\left[\langle \boldsymbol{u}_t | \boldsymbol{h}_k \rangle^2 + \langle \boldsymbol{u}_t | \boldsymbol{h}_{g(i)} \rangle^2 + 2\langle \boldsymbol{u}_t | \boldsymbol{h}_k \rangle \langle \boldsymbol{u}_t | \boldsymbol{h}_{g(i)} \rangle\right], \tag{46}$$

$$|\langle \boldsymbol{u}_t | \psi_- \rangle|^2 = \frac{1}{Z_-^2}\left[\langle \boldsymbol{u}_t | \boldsymbol{h}_k \rangle^2 + \langle \boldsymbol{u}_t | \boldsymbol{h}_{g(i)} \rangle^2 - 2\langle \boldsymbol{u}_t | \boldsymbol{h}_k \rangle \langle \boldsymbol{u}_t | \boldsymbol{h}_{g(i)} \rangle\right]. \tag{47}$$

States $|\psi_+\rangle$ and $|\psi_-\rangle$ could be generated by step 2-4 in Algorithm 6. The overlap $\langle \boldsymbol{h}_k | \boldsymbol{h}_{g(i)} \rangle$ could be estimated by Algorithm 6. The square overlap $|\langle \boldsymbol{u}_t | \psi_+ \rangle|^2$ and $|\langle \boldsymbol{u}_t | \psi_- \rangle|^2$ could be estimated by Quantum Swap Test. Thus for $|\langle \boldsymbol{u}_t | \boldsymbol{h}_{g(i)} \rangle| > \epsilon'$, one could discriminate the positive and negative of $\langle \boldsymbol{u}_t | \boldsymbol{h}_k \rangle \langle \boldsymbol{u}_t | \boldsymbol{h}_{g(i)} \rangle$ by calculate the value $Z_+^2 |\langle \boldsymbol{u}_t | \psi_+ \rangle|^2 - Z_-^2 |\langle \boldsymbol{u}_t | \psi_- \rangle|^2$. The estimation on the square overlap $|\langle \boldsymbol{u}_t | \psi_+ \rangle|^2$ and $|\langle \boldsymbol{u}_t | \psi_- \rangle|^2$ need to have the precision $\epsilon' |\langle \boldsymbol{u}_t | \boldsymbol{h}_k \rangle|/2$, which takes time $\mathcal{O}(T_{Input}\epsilon'^{-2})$. For $|\langle \boldsymbol{u}_t | \boldsymbol{h}_{g(i)} \rangle| \leq \epsilon'$, 0 is an $\epsilon'$ estimation to $\langle \boldsymbol{u}_t | \boldsymbol{h}_{g(i)} \rangle$. Since an $\epsilon'$-estimation to $|b_i|$ could be achieved by an $\epsilon'^2$-estimation to $b_i^2$ which takes time $\mathcal{O}(T_{Input}\epsilon'^{-4})$, we could derive the time complexity of estimating $b_i$ in Theorem 10.

**Theorem 10.** *There exists a quantum algorithm which could present $\epsilon$-estimation to value $b_i = sgn(u_t^{(k)})\langle \boldsymbol{u}_t | \boldsymbol{s}_i \rangle$ with probability at least $1 - \delta$ in running time $\mathcal{O}(T_{Input}\text{polylog}(d)\epsilon^{-4}\log(1/\delta))$, where $T_{Input}$ is the time complexity to generate states $|\boldsymbol{u}_t\rangle$ and $|\boldsymbol{s}_i\rangle$.*

### A.12 GENERATE THE HESSIAN MATRIX

Obtaining the Hessian matrix of a general objective function is computationally expensive. However, the calculation can be simplified in special case. For example, consider the binary classification problem with the sample set $Z = \{(\boldsymbol{x}_j, y_j) | j \in [n]\} \subset \mathbb{R}^d \times \{0, 1\}$. Consider the non-convex loss function with form: $F(\boldsymbol{x}, \boldsymbol{w}) = [h(\theta) - y]^2$, where $h : \mathbb{R} \to \mathbb{R}$ is the non-linear function such that $h(\theta) = \frac{1}{1+e^{-\theta}}$ and $\theta = \boldsymbol{w}^\top \boldsymbol{x} + b$. Denote $x_j^{(i)}$ as the $i$th component of $\boldsymbol{x}_j$. The total loss function on the set $S$ is defined as $f(\boldsymbol{w}) = \frac{1}{n}\sum_{j=1}^n F(\boldsymbol{x}_j, \boldsymbol{w}) = \frac{1}{n}\sum_{j=1}^n f_j(\boldsymbol{w})$.

There is:

$$\begin{aligned}
\frac{\partial f_j}{\partial w^{(i)}} &= 2(h(\theta_j) - y_j)\frac{\partial h(\theta_j)}{\partial w^{(i)}} \\
&= 2(h(\theta_j) - y_j)\frac{e^{-\theta_j}}{(1+e^{-\theta_j})^2}\frac{\partial \theta_j}{\partial w^{(i)}} \\
&= 2(h(\theta_j) - y_j)\frac{e^{-\theta_j}}{(1+e^{-\theta_j})^2}x_j^{(i)}
\end{aligned} \tag{48}$$

$$\frac{\partial^2 f_j}{\partial w^{(i)} \partial w^{(k)}} = 2 \frac{\partial}{\partial w^{(k)}} \left[ (h(\theta_j) - y_j) \frac{e^{-\theta_j}}{(1 + e^{-\theta_j})^2} \right] x_j^{(i)}$$

$$= 2 \left[ \frac{\partial (h(\theta_j) - y_j)}{\partial w^{(k)}} \frac{e^{-\theta_j}}{(1 + e^{-\theta_j})^2} + (h(\theta_j) - y_j) \frac{\partial}{\partial w^{(k)}} \frac{e^{-\theta_j}}{(1 + e^{-\theta_j})^2} \right] x_j^{(i)}$$

$$= 2 \left[ \frac{e^{-\theta_j}}{(1 + e^{-\theta_j})^2} x_j^{(k)} \frac{e^{-\theta_j}}{(1 + e^{-\theta_j})^2} + (h(\theta_j) - y_j) \frac{(-1) \left( e^{\theta_j} - e^{-\theta_j} \right) e^{-2\theta_j}}{(1 + e^{-\theta_j})^4} x_j^{(k)} \right] x_j^{(i)}$$

$$= 2 x_j^{(i)} x_j^{(k)} \frac{e^{-2\theta_j}}{(1 + e^{-\theta_j})^4} \left[ 1 - (h(\theta_j) - y_j) \left( e^{\theta_j} - e^{-\theta_j} \right) \right].$$

$$\tag{49}$$

Define

$$\alpha_j \equiv 2 \frac{e^{-2\theta_j}}{(1 + e^{-\theta_j})^4} \left[ 1 - (h(\theta_j) - y_j) \left( e^{\theta_j} - e^{-\theta_j} \right) \right], \tag{50}$$

we have:

$$\nabla_{\boldsymbol{w}}^2 f_j = \alpha_j \boldsymbol{x}_j \boldsymbol{x}_j^T. \tag{51}$$

Thus we could derive the Hessian matrix for the total loss function:

$$\boldsymbol{H} = \frac{1}{n} \sum_{j=1}^{n} \nabla_{\boldsymbol{w}}^2 f_j = \frac{1}{n} \sum_{j=1}^{n} \alpha_j \boldsymbol{x}_j \boldsymbol{x}_j^T. \tag{52}$$

Each term in the sum above is a rank-1 matrix, which means the rank of $\boldsymbol{H}$ is bounded by $n$. Thus, for sparse samples $\{\boldsymbol{x}_j\}_{j=1}^n$ with at most $s$ non-zero elements, we could derive the Hessian as the form $\{(i, j, h_{ij} | h_{ij} \neq 0)\}$ in time $\mathcal{O}(ns^2)$. Note that in practical case, $d$, the dimension of the sample $\boldsymbol{x}_j$, can be millions large (for example, the number of pixels on an image), while $n$, the number of sample involved in one iteration of the update, is relatively small.

