# OpenReview forum: "Quantum algorithm for finding the negative curvature direction"
_ICLR.cc/2020/Conference — Reject_

### Official Review · AnonReviewer2 · 2019-10-23
**Official Blind Review #2**

**Rating:** 6

**Review:**

This paper proposes a quantum algorithm aiming to solve the eigenvalue decomposition for the Hessian matrix in second-order optimization. The main body of this paper is the quantum singular value decomposition algorithm from Kerenidis & Prakash, 2016. The authors propose some plug-in algorithms in the context of quantum computing. Moreover, this paper does not evaluate the proposed algorithm with any comparative experiments. Personally speaking, it is better to submit the work to the physics journals.

**Experience Assessment:**

I have read many papers in this area.

**Review Assessment: Checking Correctness Of Derivations And Theory:**

I carefully checked the derivations and theory.

**Review Assessment: Checking Correctness Of Experiments:**

N/A

**Review Assessment: Thoroughness In Paper Reading:**

I read the paper thoroughly.

---

> ### Author Response · Authors · 2019-11-09
> **Response to Reviewer #2**
>
>
> Thanks for your comments. Below are some responses to the reviewer’s concern:
>
> RE_1: The main body of this paper is the quantum singular value decomposition algorithm from Kerenidis & Prakash, 2016. The authors propose some plug-in algorithms in the context of quantum computing.
>
> AS_1: This paper focuses on the efficient calculation of negative curvature direction. The contributions are two-folds: 1) a quantum algorithm to generate the target state which corresponds to the negative curvature direction, and 2) an efficient read-out algorithm to provide a classical description for the target state.
> The quantum SVE model is used for performing the quantum singular value decomposition. However, only using the quantum SVE is not enough to achieve quantum advantage because the obtained singular value does not carry the sign information, and the further read-out of the target state is another bottleneck for both this work and other quantum ML models which output state as the solution. Besides using the quantum SVE model, it is critical for us to develop several subroutines for the proposed task, for example, the PNED algorithm (Algorithm 4 in Appendix) for discriminating the sign of eigenvalue. We also develop the Complete Basis Selection Algorithm in the read-out part which aims to select a linearly independent basis from d columns in order to form the column space. Both of these are important yet non-trivial development for the proposed task.
>
> RE_2: Moreover, this paper does not evaluate the proposed algorithm with any comparative experiments.
>
> AS_2: The importance of this work is to design efficient quantum algorithms with theoretical guarantees of complexity analysis. Moreover, similar to the quantum phase estimation algorithm, the quantum SVE model used in our algorithm has the circuit depth $O(1/\epsilon)$, which makes the meaningful classical simulation unavailable (the time complexity of classical simulation grows exponentially with respect to the circuit depth).
>
> RE_3: Personally speaking, it is better to submit the work to the physics journals.
>
> AS_3: Both CS and AI communities have witnessed the fast progress of quantum computing and quantum machine learning. Many quantum algorithms have been published at top AI conferences in recent years, such as [1-4]. This paper presents a quantum algorithm for finding the negative curvature direction, which is often used in second-order algorithms for non-convex optimization. Since current deep learning models are highly non-convex, the proposed algorithm is valuable.
>
> [1]. Kapoor, Ashish, Nathan Wiebe, and Krysta Svore. "Quantum perceptron models." NeurIPS 2016.
> [2]. Aaronson, Scott, et al. "Online learning of quantum states." NeurIPS 2018.
> [3]. Li, Tongyang, Shouvanik Chakrabarti, and Xiaodi Wu. "Sublinear quantum algorithms for training linear and kernel-based classifiers." ICML 2019.
> [4]. Kerenidis, Iordanis, et al. "q-means: A quantum algorithm for unsupervised machine learning." NeurIPS 2019.

---

### Official Review · AnonReviewer3 · 2019-10-24
**Official Blind Review #3**

**Rating:** 6

**Review:**

This paper proposes a new quantum algorithm which can efficiently estimate the eigen vectors of a Hessian matrix with negative eigenvalue.

Decision

I would vote for a weak accept although this is somewhat an educated guess. Finding efficient ways to estimate eigen-vectors of the Hessian would have a dramatic impact on second-order optimization techniques (neglecting the practical implications of it being a quantum algorithm). Since the paper is submitted to a machine learning conference I believe more efforts should be done to make the paper accessible to researchers unfamiliar to quantum computing.

Comments

I appreciate the clarification of the notation at section 2.1, however many exotic notation for machine learning researchers are presented before this section without any reference to 2.1. The operation `\propto` (latex) is never explained. My understanding is that it is the update of a quantum state?

The section 2.2 should give a brief overview of why both techniques will be useful for the many parts of the contributed algorithms.

I could not find definition of T_H anywhere.

**Experience Assessment:**

I do not know much about this area.

**Review Assessment: Checking Correctness Of Derivations And Theory:**

I did not assess the derivations or theory.

**Review Assessment: Checking Correctness Of Experiments:**

N/A

**Review Assessment: Thoroughness In Paper Reading:**

I read the paper at least twice and used my best judgement in assessing the paper.

---

> ### Author Response · Authors · 2019-11-09
> **Response to Reviewer #3**
>
>
> Thanks for your comments. We would like to clarify our contributions:
>
> This paper focuses on the efficient calculation of negative curvature direction. The contributions are two-folds: 1) a quantum algorithm to generate the target state which corresponds to the negative curvature direction, and 2) an efficient read-out algorithm to provide a classical description for the target state.
> The quantum SVE model is used for performing the quantum singular value decomposition. However, only using the quantum SVE is not enough to achieve quantum advantage because the obtained singular value does not carry the sign information, and the further read-out of the target state is another bottleneck for both this work and other quantum ML models which output state as the solution. Besides using the quantum SVE model, it is critical for us to develop several subroutines for the proposed task, for example, the PNED algorithm (Algorithm 4 in Appendix) for discriminating the sign of eigenvalue. We also develop the Complete Basis Selection Algorithm in the read-out part which aims to select a linearly independent basis from d columns in order to form the column space. Both of these are important yet non-trivial development for the proposed task.
>
> Below are some responses to the reviewer’s concern:
>
> RE_1: The operation `\propto` (latex) is never explained. My understanding is that it is the update of a quantum state?
>
> AS_1: Thanks and revised.
>
> RE_2: I could not find definition of T_H anywhere.
>
> AS_2: Thanks. T_H is defined as the time complexity of constructing oracles $U_H$ and $V_H$ in Section 2.2 on page 4.

---

### Official Review · AnonReviewer1 · 2019-10-29
**Official Blind Review #1**

**Rating:** 3

**Review:**

I am not familiar with the quantum algorithm literature, hence I cannot judge the novelty of the proposed algorithm. Basically, this is a quantum algorithm for computing the smallest negative eigenvalue (and the associated eigenvector) of square matrices. It is worth noting that the algorithm is built on top of the existing SVE model. The authors don't compare this method with any existing work in quantum singular value transformation and decomposition. A throughout discussion and comparison should be made in Section 1.1

Despite the title, I can't find a strong connection between the proposed algorithm and machine learning. Given a loss function f and the parameters x, the paper doesn't specify how to construct the quantum state in the step 2 of Algorithm 2. Is there a cheap way to construct the quantum state from (f, x)? If not, and if one has to compute the Hessian matrix H from (f, x), then the complexity of this pre-processing step will be on the order of O(d^2), which already makes the quantum efficiency meaningless.

Escaping saddle point is usually an iterative process, because the curvature varies across from place to place. The paper focuses on an individual step of a single iteration (i.e. finding the negative curvature), but it doesn't mention how would the iterative process look like.

Finally, although the quantum complexity is poly-log in dimension d, it has a terrible dependence on the rank r and the Frobenius norm of the Hessian matrix. The potential issues of such dependencies should be discussed.





**Experience Assessment:**

I do not know much about this area.

**Review Assessment: Checking Correctness Of Derivations And Theory:**

I assessed the sensibility of the derivations and theory.

**Review Assessment: Checking Correctness Of Experiments:**

N/A

**Review Assessment: Thoroughness In Paper Reading:**

I read the paper at least twice and used my best judgement in assessing the paper.

---

> ### Author Response · Authors · 2019-11-09
> **Response to Reviewer #1 (2/2)**
>
>
> RE_4: If not, and if one has to compute the Hessian matrix H from (f, x), then the complexity of this pre-processing step will be on the order of O(d^2), which already makes the quantum efficiency meaningless.
>
> AS_4: For the special class of objective function, the Hessian can be derived efficiently, for example, the scenario of binary classification with one neural layer and sigmoid non-linear function. We provide related details in Appendix (A. 12). For sample set which contains n d-dimensional s-sparse samples, the time complexity of calculating the Hessian is $O(ns^2)$. Previous work Kerenidis & Prakash, 2016 presents a technique about constructing the quantum oracle model by a $O(wlog^2 d)$ time complexity procedure, where d is the dimension and w is the number of none zero elements in Hessian matrix.
>
> RE_5: Finally, although the quantum complexity is poly-log in dimension d, it has a terrible dependence on the rank r and the Frobenius norm of the Hessian matrix.
>
> AS_5: Our algorithm has a quasi-polynomial order with respect to the rank r (the Frobenius norm of the Hessian matrix can be bounded by $O(\sqrt{r})$). The recent work [4] shows the low-rank property of the Hessian matrix for the neural networks, and so such an order on the rank of the Hessian is acceptable.
>
>
> [1]. Seth Lloyd, Masoud Mohseni, and Patrick Rebentrost. Quantum principal component analysis. Nature Physics, 10(9):631, 2014.
> [2]. Patrick Rebentrost, Adrian Steffens, Iman Marvian, and Seth Lloyd. Quantum singular-value de- composition of nonsparse low-rank matrices. Physical review A, 97(1):012327, 2018.
> [3]. Scott Aaronson. Quantum machine learning algorithms : Read the fine print. 2015.
> [4]. Guy Gur-Ari, Daniel A Roberts, and Ethan Dyer. Gradient descent happens in a tiny subspace. arXiv preprint arXiv:1812.04754, 2018.

---

> ### Author Response · Authors · 2019-11-09
> **Response to Reviewer #1 (1/2)**
>
>
> Thanks for your comments. We would like to clarify our contributions in this work:
>
> This paper focuses on the efficient calculation of negative curvature direction. The contributions are two-folds: 1) a quantum algorithm to generate the target state which corresponds to the negative curvature direction, and 2) an efficient read-out algorithm to provide a classical description for the target state.
> The quantum SVE model is used for performing the quantum singular value decomposition. However, only using the quantum SVE is not enough to achieve quantum advantage because the obtained singular value does not carry the sign information, and the further read-out of the target state is another bottleneck for both this work and other quantum ML models which output state as the solution. Besides using the quantum SVE model, it is critical for us to develop several subroutines for the proposed task, for example, the PNED algorithm (Algorithm 4 in Appendix) for discriminating the sign of eigenvalue. We also develop the Complete Basis Selection Algorithm in the read-out part which aims to select a linearly independent basis from d columns in order to form the column space. Both of these are important yet non-trivial development for the proposed task.
>
> Below are some responses to the reviewer’s concern:
>
> RE_1: The authors don't compare this method with any existing work in quantum singular value transformation and decomposition.
>
> AS_1: We have added notes about the quantum algorithms for the related linear algebra problems in the revision. There are some proposed quantum algorithms for problems in the related linear algebra field. For example, given copies to quantum state $\rho=\sum_{i,j=1}^d x_{ij} |i\rangle\langle j|$, where $x_{ij}$ is the $i,j$-th element of $d \times d$ matrix $X$ with eigen-decomposition $X=\sum_{i=1}^{rank(X)} \lambda_i  {u}_i  {u}_i$, previous quantum PCA algorithm [1] could perform the mapping $\sum_j \beta_j | {u}_j\rangle \rightarrow \sum_j \beta_j | {u}_j\rangle |\tilde{\lambda}_j\rangle$ in time $O(polylog(d)\epsilon^{-3})$, which implies one sort of quantum SVD. However, the quantum PCA model use the density matrix $\rho$ to store the information of concerned matrix $X$, which implicitly assumes conditions $\|X\|_F=1$ and $\lambda_{\min}(X) \ge 0$. Another quantum SVD algorithm [2] shows an efficient method to estimate the value $\lambda_j /d$ with error $\epsilon$ in time $O(\epsilon^{-3})$, for $d \times d$ matrix $X$ with eigenvalues $\{\lambda_j\}_{j=1}^d$. However, this model only suits the case when $\lambda_j/d$ is relatively large, and it would take time $O(d^3 \epsilon^{-3})$ to produce $\epsilon$-estimation on eigenvalues.
> Moreover, both methods do not study the classical read-out of the output state, which generally takes time at least $O(d)$ for $d$-dimensional state, offsetting the claimed quantum speed-up [3]. It is thus critical to design fast read-out algorithms to achieve quantum advantage for the whole task, which is usually challenging.
>
> RE_2: Despite the title, I can't find a strong connection between the proposed algorithm and machine learning.
>
> AS_2: This work presents a quantum algorithm for finding the negative curvature direction, which is often used in many second-order algorithms for non-convex optimization. Since current deep learning models are highly non-convex, our algorithm is valuable.
>
> RE_3: Given a loss function f and the parameters x, the paper doesn't specify how to construct the quantum state in the step 2 of Algorithm 2. Is there a cheap way to construct the quantum state from (f, x)?
>
> AS_3: We place the construction of state in the step 2 of Algorithm 2 in Appendix (A. 4) due to the limit of page length. Specifically, the quantum state could be prepared by applying oracles V_H and U_H on the initial $|0\rangle|0\rangle$ state, each for one time:
>
> $|0\rangle|0\rangle \stackrel{V_H}{\longrightarrow} \frac{1}{\| {H}\|_F} \sum_{i=1}^d\| {h}_i\||i\rangle|0\rangle \stackrel{U_H}{\longrightarrow} \frac{1}{\| {H}\|_F} \sum_{i=1}^d \sum_{j=1}^d h_{ij} |i\rangle|j\rangle = \frac{1}{\| {H}\|_F}\sum_{k=1}^{r} \lambda_k | {u}_k\rangle | {u}_k\rangle$.

---

### Decision · Program_Chairs · 2019-12-19

**Decision:**

Reject

**Comment:**

There was some support for the ideas presented, but this paper was on the borderline, and ultimately not able to be accepted for publication at ICLR.

Concerns raised included level of novelty, and clarity of the exposition to an ML audience.